# Combining commercial microwave links and weather radar for classification of dry snow and rainfall

Erlend Øydvin[1], Renaud Gaban[1, 5], Jafet Andersson[2], Remco (C. Z.) van de Beek[2], Mareile Astrid Wolff[1, 5], Nils-Otto Kitterød[4], Christian Chwala[3], and Vegard Nilsen[1]

[1]Faculty of Science and Technology, Norwegian University of Life Sciences, Ås, Norway
[2]Swedish Meteorological and Hydrological Institute (SMHI), 601 76 Norrköping, Sweden
[3]Institute of Meteorology and Climate Research, Karlsruhe Institute of Technology, Campus Alpin, Garmisch-Partenkirchen, Germany
[4]Faculty of Environmental Sciences and Natural Resource Management, Norwegian University of Life Sciences, Ås, Norway
[5]Norwegian Meteorological Institute, Oslo, Norway

**Correspondence:** Erlend Øydvin (erlend.oydvin@nmbu.no)

**Abstract.** Differentiating between snow and rainfall is crucial for hydrological modeling and understanding. Commercial Microwave Links (CMLs) can provide rainfall estimates for liquid precipitation, but show minimal signal attenuation during dry snow events, causing the CML time series during these periods to resemble non-precipitation periods. Weather radars can detect precipitation also for dry snow, yet they struggle to accurately differentiate between precipitation types. This study introduces a new approach to improve rainfall and dry snow classification by combining weather radar precipitation detection with CML signal attenuation. Specifically, events where the radar detects precipitation, but the CML does not, are classified as dry snow. As a reference method, we use weather radar with the precipitation type identified by the dew point temperature at the CML location. Both methods were evaluated using measurements from disdrometers located within 8 km of a CML, taken as ground truth. The analysis used data from Norway, including 550 CMLs in December 2021 and 435 CMLs in June 2022. Our results show that the use of CMLs can improve the classification of dry snow and rainfall, presenting an advantage over the reference method. In addition, our research provides valuable insight into how precipitation at temperatures around 0 °C, such as sleet or wet snow, can affect CMLs, contributing to a better understanding of CML applications in colder climates.

## 1 Introduction

The precipitation phase is crucial for hydrological processes in cold regions (Loth et al., 1993). Understanding the type of precipitation aids in applications such as adjusting rain gauges for wind undercatch (Kochendorfer et al., 2022) and modeling hydrological responses such as flooding. Moreover, specific precipitation conditions, like freezing rain, can disrupt power lines and impede traffic, and snow may cause transportation blockages. Rain-on-snow events have also been associated with significant flooding (McCabe et al., 2007), and with slush avalanches (Hestnes, 1985).

The formation of precipitation is a complex process. In the high and mid-latitudes, most precipitation originates from mixed-phase or cold clouds, that is clouds containing ice (Stewart et al., 2015). The ice crystals grow in size and mass through different micro-physical mechanisms such as vapor deposition and riming, until they reach a sufficient mass to sediment out of the cloud

base (Stewart, 1992). A necessary condition for a cloud to generate solid-phase precipitation is that its temperature is negative. Unless the temperature of the layer of atmosphere underneath the cloud remains below 0 °C, the precipitating ice will start melting before reaching the ground (Lamb and Verlinde, 2011). The melting process is not instantaneous and an ice particle can fall hundreds of meters before melting completely. Therefore, originally solid precipitation can reach the ground in any intermediate state between solid and liquid, depending on the elevation of the 0 °C isotherm (Paulson and Al-Mreri, 2011; Harpold et al., 2017). The melting process is also influenced by other elements of the atmospheric conditions. Specifically, the stability of the atmosphere and the atmospheric humidity profile have a significant influence because the liquid water formed from the melting of ice will tend to evaporate in dry conditions, cooling the atmosphere in turn and hampering further melting (Harder and Pomeroy, 2013). Atmospheric conditions that determine the precipitation phase can change relatively quickly. For instance, a study by Marks et al. (2013) observed a significant increase of the elevation of the melting layer within the same precipitation event. Determining the precipitation phase at the ground level is therefore difficult and models predicting the precipitation phase typically need to be calibrated and validated against measurements (Harpold et al., 2017).

There are several ways of determining the precipitation phase using ground-based observations. For example, Marks et al. (2013) suggested using a combination of a tipping bucket rain gauge and a heated weighing gauge. During rain, the devices record similar amounts, but when it snows, the snow clogs the funnel of the tipping bucket, leaving only the weighing gauge to record precipitation. Other studies, such as Matsuo et al. (1981), used human observers to directly observe the precipitation phase. More advanced methods include using weather radars, especially with dual polarization, to estimate and classify precipitation phase (Grazioli et al., 2015; Chandrasekar et al., 2013). Disdrometers also estimate precipitation phase based on the physical properties of hydrometeors (size and fall velocity), with semi-empirical knowledge of how these properties vary with type (Löffler-Mang and Joss, 2000; Yuter et al., 2006). However, each method has its own limitations. Rain gauges, providing point measurements, have limited spatial representation and can be affected by wind-induced errors (Førland et al., 1996; Nešpor and Sevruk, 1999; Kochendorfer et al., 2022; Wolff et al., 2015). Human observations can be subjective and are not suitable for continuous high-rate monitoring. Weather radars can suffer from beam blockage (Berne and Krajewski, 2013), overshooting, and still have difficulties linking the estimated precipitation type to ground measurements (Harpold et al., 2017; Elmore, 2011). Like rain gauges, disdrometers are limited in spatial representation and can experience errors such as splashing of drops against nearby structures, drops falling on the edge of the measuring area, and wind altering the drop trajectories (Friedrich et al., 2013).

Precipitation phase estimation often employs a temperature model. This involves modeling the rain-snow transition based on temperature using, for instance, a single temperature threshold to separate rain and snow, or two thresholds that define mixed precipitation in between the two thresholds (Kienzle, 2008). Jennings et al. (2018) found that, when using a single temperature threshold, the threshold separating rain from snow varies geographically, ranging from -0.4 to 2.4 for most stations and with colder thresholds near the coast and warmer thresholds in the mountains. They also found that models incorporating humidity performed better than models considering air temperature alone, which is confirmed by other studies (Matsuo et al., 1981). Although some studies do not observe any benefit of including humidity (Leroux et al., 2023), humidity is thought to be an important parameter since the atmospheric moisture level affects the melting and evaporating precipitation, influencing whether

precipitation reaches the ground as solid or liquid (Kuhn, 1987). One common measure combining humidity and temperature is the dew point temperature, which is the temperature at which the air would become saturated with water vapor at constant pressure and moisture content (Lawrence, 2005). In a dry atmosphere, the dew point temperature is significantly lower than the

air temperature. In these conditions, melting is not favored and snow can be observed at positive air temperatures. Conversely, the dew point and the air temperatures are equal if the air is saturated, and solid precipitation will more likely have melted before reaching the ground if the air temperature is above 0 °C (Feiccabrino, 2020; Harder and Pomeroy, 2013, 2014). However, a large degree of uncertainty in precipitation type classification remains even when temperature and humidity are combined. Harpold et al. (2017) suggest that current phase transition models are too simple to capture the process, especially in complex

terrain. They suggest, for instance, to improve this by better use of other atmospheric information and enhancing the validation network with ground measurements such as disdrometers.

Commercial microwave links (CMLs) are radio links between radio communication towers. In the mid-2000s, it was demonstrated by Messer et al. (2006) and Leijnse et al. (2007) that CMLs can be used to estimate rainfall. This is due to the relationship between signal attenuation and rainfall intensity. At around 30 GHz, the relation is close to linear, making it easier to estimate

the average rainfall intensity along the CML path. Among other applications, CMLs have been used to estimate countrywide rainfall (Graf et al., 2020; Overeem et al., 2016), transboundary rainfall fields (Blettner et al., 2023) and CMLs have proven useful for estimating runoff in urban hydrology (Pastorek et al., 2023). A crucial step in CML rainfall estimation is the detection of rainfall, often called wet periods, in the CML time series. There are several ways of doing this, for instance by classifying a period as wet when the standard deviation of a moving window is larger than a predefined threshold (Schleiss et al., 2013;

Graf et al., 2020), by using pre-trained classification models (Polz et al., 2020; Øydvin et al., 2024) or by including information from nearby CMLs (Overeem et al., 2011). It is also possible to use weather radar to estimate the CML wet period as done in Overeem et al. (2016).

Classification of precipitation types other than rain using CMLs has previously been investigated by Cherkassky et al. (2014). The authors used the fact that snow, sleet (defined as a mixture of snow and rain), and rainfall are affected differently

by different CML frequencies. Thus, by using three CMLs operating at different frequencies in the same area, they were able to distinguish periods of sleet and rainfall, albeit only for two precipitation events, each lasting three days. Ostrometzky et al. (2015) expanded on this study by using four CMLs operating at different frequencies and clustered at a single path to estimate the precipitation amounts generated by rainfall and sleet. The study investigated four precipitation events lasting a total of 16 days. A limitation of both of these studies is that they focus on a low number of CMLs over a few days. It is not known

how well these methods generalize to longer time series and larger CML networks. Other studies have focused on how CMLs are affected by colder climates. Hansryd et al. (2010) reported that heavy snowfall caused minimal signal attenuation, while a mix of rain and snow, caused higher signal attenuation. van Leth et al. (2018) observed that during an event with mixed precipitation, the CMLs experienced a strong signal attenuation which persisted for about 10 minutes after the precipitation event, possibly due to the melting of snow off from the antenna cover. Graf et al. (2020) and Overeem et al. (2016) reported

that CMLs tend to overestimate the precipitation amount during winter months. Both attributed this overestimation to melting snow, which is known to cause up to four times the attenuation compared to rainfall, depending on the mixture of snow and

rain (Paulson and Al-Mreri, 2011). Dry snow, on the other hand, is known to cause signal attenuation so low that it cannot be detected by CMLs (Pu et al., 2020; Paulson and Al-Mreri, 2011; Hansryd et al., 2010).

In this study, we explore the viability of classifying dry snow by exploiting the fact that dry snow causes unnoticeable attenuation in the CML data. This is done by first using the weather radar to detect precipitation and then classifying the precipitation type based on whether the CML detects rainfall or not. We compare these estimates to ground truth observations from disdrometers operated by the road authorities in Norway, as well as a reference method that uses weather radar and dew point temperature to estimate the precipitation phase.

## 2    Methods

### 2.1    CML data

The CML dataset was provided by Ericsson and consists of 2777 CMLs spread out across Norway. Each CML records the transmitted and received signal strength every minute for data from two months: December 2021 and June 2022. The CML signal attenuation, often called total loss (TL), was computed by subtracting the received signal strength from the transmitted signal strength. In our dataset, there were some outliers where the transmitted signal strength was less than -50 dBm. These signals produced negative TL values, likely due to recording errors. We opted to completely remove CMLs with transmitted signal strength less than -50 dBm such that the remaining CMLs did not have any negative TL values. We also removed CMLs with more than 15% missing values. This resulted in 2179 CMLs for the summer dataset and 2345 CMLs for the winter dataset. Next, as suggested by Graf et al. (2020), we removed erratic CMLs where the 5 hours moving window standard deviation exceeded the threshold of 2 dBm more than 10 % of the month and noisy CMLs where the 1-hour moving window standard deviation exceeded the threshold of 0.8 dBm more than 33 % of the month. Then CML derived rain rates were estimated using the *pycomlink* software (Chwala et al., 2023) and a similar workflow as described in Graf et al. (2020), Blettner et al. (2023) and Polz et al. (2020). For the classification of rainy periods, we used the convolutional neural network (CNN) developed by Polz et al. (2020), as this model was trained on hourly data and this study also evaluates estimates at an hourly resolution. The baseline was estimated by using the average signal attenuation 5 minutes before a rainy period. Water droplets forming on the antenna cover cause additional attenuation, which we accounted for by using the semi-empirical wet antenna attenuation model proposed by Leijnse et al. (2008), with the refined parameters suggested by Graf et al. (2020). Finally, the rainfall rate was computed using the k-R relation, with parameters defined by ITU (2005).

### 2.2    Radar data

Weather radar data for Norway was downloaded from THREDDS (2024), a data hosting platform for gridded meteorological data run by the Norwegian Meteorological Institute. The radar product is developed from 12 weather radars in Norway. These radars are combined using a Constant Altitude Plan Position Indicator (CAPPI). The final result is a grid with a spatial resolution of 1 km by 1 km and a temporal resolution of 5 minutes. Sea clutter and other large peaks in the data are removed, and

ground clutter is identified and corrected using surrounding data. The radar reflectivity (Z [dBz]) is converted to precipitation rates (R [mm/h ]) using the Marshall-Palmer relation (Marshall and Palmer, 1948). Radar precipitation rates were estimated along all CMLs using the weighted grid approach provided by *pycomlink*. Then, in line with Polz et al. (2020), time steps with weather radar rainfall rates above 0.1 mm/h were considered rainy.

## 2.3 Disdrometer data and co-located CMLs

As ground truth for the precipitation type, we used disdrometer data from the Norwegian road authorities. They use two types of disdrometers, namely the OTT Parsivel and OTT Parsivel[2]. The disdrometer data was downloaded from Frost (2024), a data hosting platform for meteorological observations run by the Norwegian Meteorological Institute. This dataset also contains precipitation type observations from other sensor types, such as the Vaisala PWD12/31 and DRD11A. Using a registry provided by the road authorities, we selected time series that were generated using the OTT Parsivel and OTT Parsivel[2]. This was done to ensure a more controlled comparison between sensors. The disdrometers are placed at least 4 meters above the road at automated meteorological stations located along the roads in Norway and provide an estimate of the precipitation type every 10 minutes. The instruments classify precipitation as light rain, rain, snow, and hail. No precipitation, or dry weather, is denoted *dry* in the following. We simplified the classification by merging the classes light rain and rain since they should appear similar in the CML and radar. Additionally, since hail events were rare in the dataset (less than 0.01%) and were not of interest to our study, we set time steps where the disdrometer recorded hail to dry. This adjustment resulted in a negligible error while simplifying the methodology. Thus from the simplifications, the disdrometers only report 3 classes: dry, rain and snow. Pairs of CMLs and disdrometers within 8 km of each other were identified using methods from poligrain (2024) and CMLs longer than 8 km were removed. The threshold of 8 km was a trade-off between getting a large number of CML-disdrometer pairs, while still maintaining a good correlation between the pairs. This resulted in 376 CMLs and 83 co-located disdrometers for December 2021 and 304 CMLs and 60 co-located disdrometers for June 2020. We refer to these as the winter and summer datasets respectively. The CMLs used in the study had signal quantization equal to 0.3 dBm, lengths ranging from 0.4 to 8 km, and operated at frequencies between 10 to 40 GHz, with the majority (90 percent) operating above 18 GHz (Fig. 1).

### 2.3.1 Temperature, humidity and dew point temperature

Temperature and humidity data were downloaded from THREDDS (2024). The temperature data is a downscaled version of ERA5 data that is combined with ground observations on a 1 km grid with a temporal resolution of 1 hour (MET, 2024; Lussana et al., 2021, 2019). Lussana et al. (2019) provide an extensive analysis of the temperature dataset, and the uncertainty of the data depends on several factors like distance to the closest observation station, terrain complexity and model assumptions. For each CML, we extracted the temperature and humidity at the midpoint of the CML. To account for air humidity we calculated the dew point temperature using the approximated relation with air temperature provided by Lawrence (2005), given as

$$T_d = T_a - ((100 - RH)/5), \tag{1}$$

where $T_d$ (°C) is the dew point temperature, $T_a$ (°C) is the air temperature and $RH$ (%) is the relative humidity.

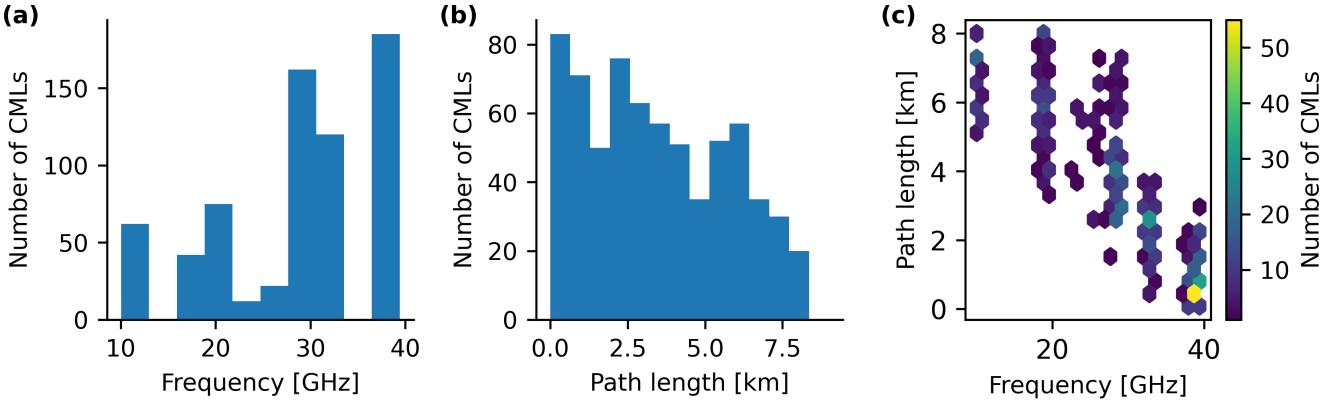

**Figure 1.** CML characteristics for the summer and winter dataset. **(a)** Distribution of CML frequencies. **(b)** Distribution of path lengths. **(c)** Relationship between path length, frequency, and number of CMLs.

## 2.4 Classification of rainy and snowy hours using the disdrometers, the CML-radar (CR) method and the radar-temperature (RT) method

The weather radar (5-minute resolution), temperature model (60-minute resolution), CML (1-minute resolution), and disdrometer (10-minute resolution) provide estimates at different locations and with varying time resolutions. This lack of synchronization could potentially lead to erroneous comparisons. To address this issue, we aggregated the precipitation type estimates to hourly intervals to help smooth out these differences.

The "CML-radar" (CR) method uses CML wet periods, as predicted by the wet-dry estimation method, to classify precipitation as rain. Next, since the CMLs are not noticeably attenuated by dry snow, radar precipitation without corresponding CML precipitation is classified as snow. To aggregate the CR estimates to hourly resolution, we classified hours as rainy if the CML rainfall classification algorithm recorded any rainfall, snowy if the radar estimated any precipitation but the CML did not estimate rainfall, and dry otherwise. The "radar-temperature" (RT) method uses surface temperature and weather radar to determine precipitation occurrence and precipitation type. As recommended by Harder and Pomeroy (2014), humidity is accounted for by using the dew point temperature. The RT method then works by estimating the average precipitation along the CML and classifying precipitation above a dew point temperature threshold as rain and below the threshold as snow. The dew point temperature was evaluated at the pixel closest to the center of the CML and was, in line with Marks et al. (2013), set to 0 °C. Combining humidity-corrected temperature models with weather radar is a common method and is often used as a reference method (Casellas et al., 2021; Saltikoff et al., 2015; Gjertsen and Ødegaard, 2005). To aggregate the RT estimates to hourly resolution we classified hours as dry if the radar observed no precipitation. If the radar observed precipitation, we classified the hours as rainy or snowy based on whether the dew point temperature was above or below 0 °C, respectively. For the disdrometers, we classified hours as rainy if the disdrometer recorded any rain during that hour, snowy if the disdrometer recorded snow but no rain, and dry otherwise.

**Table 1.** Confusion matrix for binary classification. The positive class typically represents snow, rain, or both, while the negative class represents all other conditions not included in the positive class.

|  | **Predicted: Negative** | **Predicted: Positive** |
|---|---|---|
| **Ground truth: Negative** | True Negative (TN) | False Positive (FP) |
| **Ground truth: Positive** | False Negative (FN) | True Positive (TP) |

The drawback of the aggregation method is that we overestimate the number of rainfall events, introducing more uncertainties in the results. However, we consider this an acceptable trade-off as it allows for a more consistent analysis. The effects of the aggregation method are evaluated further in the discussion.

## 2.5 Metrics

In this study, we will consider both binary classification (for instance, rain and not rain) and multiclass classification (rain, snow and dry). A common tool for visualizing the performance of a classification algorithm is to use a confusion matrix. It compares the ground truth labels with the predicted labels, allowing us to understand the types of errors made by the classifier. In this study, the classifier is either the CR or RT method and the ground truth is the disdrometers. In the binary case, the labels are categorized as either negative or positive (Table 1). Using this confusion matrix, several metrics can be defined. We will first consider the the accuracy score, which relates the proportion of true results to the total number of cases examined such that

$$\text{Accuracy} = \frac{\text{TN} + \text{TP}}{\text{TN} + \text{TP} + \text{FP} + \text{FN}}. \tag{2}$$

Because of its simplicity, accuracy is a widely used metric, but it may not always be the best indicator of a classifier's performance, especially in cases of imbalanced datasets. To better understand the performance of the classifiers it is therefore common to consider other metrics as well. The precision score focuses on the classifiers' positive predictions and provides the ratio between the false and true positive predictions, such that

$$\text{Precision} = \frac{\text{TP}}{\text{TP} + \text{FP}}. \tag{3}$$

It is also common to consider the recall score, which gives the proportion of reference positive cases predicted by the classifier

$$\text{Recall} = \frac{\text{TP}}{\text{TP} + \text{FN}}. \tag{4}$$

Since any classifier typically must strike a balance between false positives and false negatives, improving the precision comes at the expense of the recall. To consider both recall and precision, the F1 score estimates the harmonic mean of the precision and recall, combining these two metrics such that

$$\text{F1} = 2 \times \frac{\text{Precision} \times \text{Recall}}{\text{Precision} + \text{Recall}} = \frac{2\text{TP}}{2\text{TP} + \text{FN} + \text{FP}}. \tag{5}$$

We also use the Matthews correlation coefficient (MCC) to quantify the performance of the classification methods. The MCC is a metric that has been shown to outperform the accuracy and F1 score on imbalanced datasets, by indicating a good correlation only when the classifier performs well on both positive and negative cases (Chicco and Jurman, 2020). The MCC is given by

$$MCC = \frac{TP \cdot TN - FP \cdot FN}{\sqrt{(TP+FP) \cdot (TP+FN) \cdot (TN+FP) \cdot (TN+FN)}}. \tag{6}$$

The accuracy, precision, recall and F1 score reach their best value at 1 and worst score at 0. The MCC score of 1 indicates a perfect prediction, 0 indicates a random guess and -1 is a perfect inverse prediction.

In the multiclass case, the hourly predicted estimates compared to the hourly ground truth can be summarized in a multiclass confusion matrix, see Table 2. The accuracy score can be extended to the multiclass case as follows

**Table 2.** Multiclass confusion matrix of the ground truth (disdrometer) and predictions (CR and RT method).

|  | Predicted: Dry | Predicted: Rain | Predicted: Snow |
|---|---|---|---|
| **Ground truth: Dry** | True Dry | False Rain | False Snow |
| **Ground truth: Rain** | False Dry | True Rain | False Snow |
| **Ground truth: Snow** | False Dry | False Rain | True Snow |

$$Accuracy = \frac{True\ dry + True\ rain + True\ snow}{Total\ number\ of\ instances}. \tag{7}$$

The precision, recall and F1 scores are slightly more complicated to generalize, as they involve selecting specific parts of the binary confusion matrix. In this work, we generalized the precision, recall and F1 scores using macro averaging, which works by evaluating each class (dry, rain snow) individually and then averaging them into a single number.

Following Gorodkin (2004), the MCC metric can be generalized to $n$ classes, providing a balanced metric that uses the full confusion matrix (Jurman et al., 2012). The MCC between the reference $x$ and the predictions $y$ for the multiclass case is given by

$$MCC = \frac{cov(X,Y)}{\sqrt{cov(X,X) \cdot cov(Y,Y)}}, \tag{8}$$

where

$$cov(X,Y) = \frac{1}{N} \sum_{k=1}^{N} cov(X_k, Y_k), \tag{9}$$

where $N$ is the number of classes and $cov(X_k, Y_k)$ is the covariance between the reference $X_k$ and the predictions $Y_k$ for class $k$. All metrics were computed using *scikit-learn* (Pedregosa et al., 2011).

## 3 Results

### 3.1 Case study

To illustrate the RT and CR method, we first focus on a case study performed over a large area in mid-Norway, covering 36 CML-disdrometer pairs over two days from the 18th to the 20th of December 2021 (Fig. 2). The area spans from the Trondheim fjord and Norwegian coast in the north and west, to the Dovre mountain in the south. For each CML we counted the number of snowy and rainy hours estimated by the RT method **(a, d)**, CR method **(b, e)** and the nearby disdrometer **(c, f)**. The total snowy and rainy hours were then interpolated using inverse distance weighting, with the midpoint of the CML indicating its position. We can observe that at the location of CML1 and CML2 (mountainous areas), the RT and CR method estimates more snow and less rain compared to the disdrometers. At the location of CML3 the RT method estimates more snow and less rain compared to the CR method and disdrometers.

Examining the time series for CML1 and CML2 (Fig. 3 and Fig. 4), we observe that before 18:00 on December 18th, the dew point temperature **(b)** is mostly above 0 °C, and the CML signal level fluctuates markedly **(c)**, especially during periods when the weather radar estimates rainfall. After 18:00, as the temperature drops well below 0 °C, the CML signal loss shows a different pattern with much less fluctuation. At the same time, the CR and RT methods estimate snow, while the disdrometer shifts between rain and snow. Before 18:00, the RT method estimates some rainfall while the CR method estimates snow, possibly because the rainfall event was not strong enough for the CML wet detection algorithm to identify it as rainfall.

The time series for CML3 indicates a different climatological pattern compared to CML1 and CML2, with temperatures above 0 °C before 18:00 on December 18th and hovering around 0 °C afterward (Fig. 5). Observing the CML time series, we see that the signal loss fluctuates during all time steps when the radar estimates precipitation. The disdrometer primarily indicates rainfall, except for a period from 00:00 to 09:00 on December 19th, when it shows a mixture of rain and snow. In contrast, the RT method indicates a longer snowy period, lasting from 19:00 on the 18th to 14:00 on the 19th. The CR method primarily estimates rainfall but identifies short snowy periods around the disdrometers snow period, and estimates rainfall when the disdrometer indicates snow.

### 3.2 Overview of the data as a function of dew point temperature

To get an overview of the full dataset used in the study, we counted the total number of rainy and snowy hours estimated by the disdrometers, RT method, and CR method for dew point temperature intervals of 1 °C between -20 and 20 °C (Fig. 6). Our data covers the full temperature range, with most observations concentrated between -10 and 10 °C. The disdrometers record snow mainly below 0 °C **(a)**, with some snow events slightly above 0 °C. Most rainy hours recorded by the disdrometers occur above 0 °C, but there are also a substantial number of rainy hours recorded below 0 °C. The CR method also estimates rainy hours below 0 °C, though less frequently than the disdrometers, and it estimates a significant number of snowy hours above 0 °C **(c)**. The RT method, by definition, predicts rainy hours above 0 °C and snowy hours below 0 °C **(b)**. Additionally, we observe that the RT method estimates fewer rainy hours than the CR method, even above 0 °C.

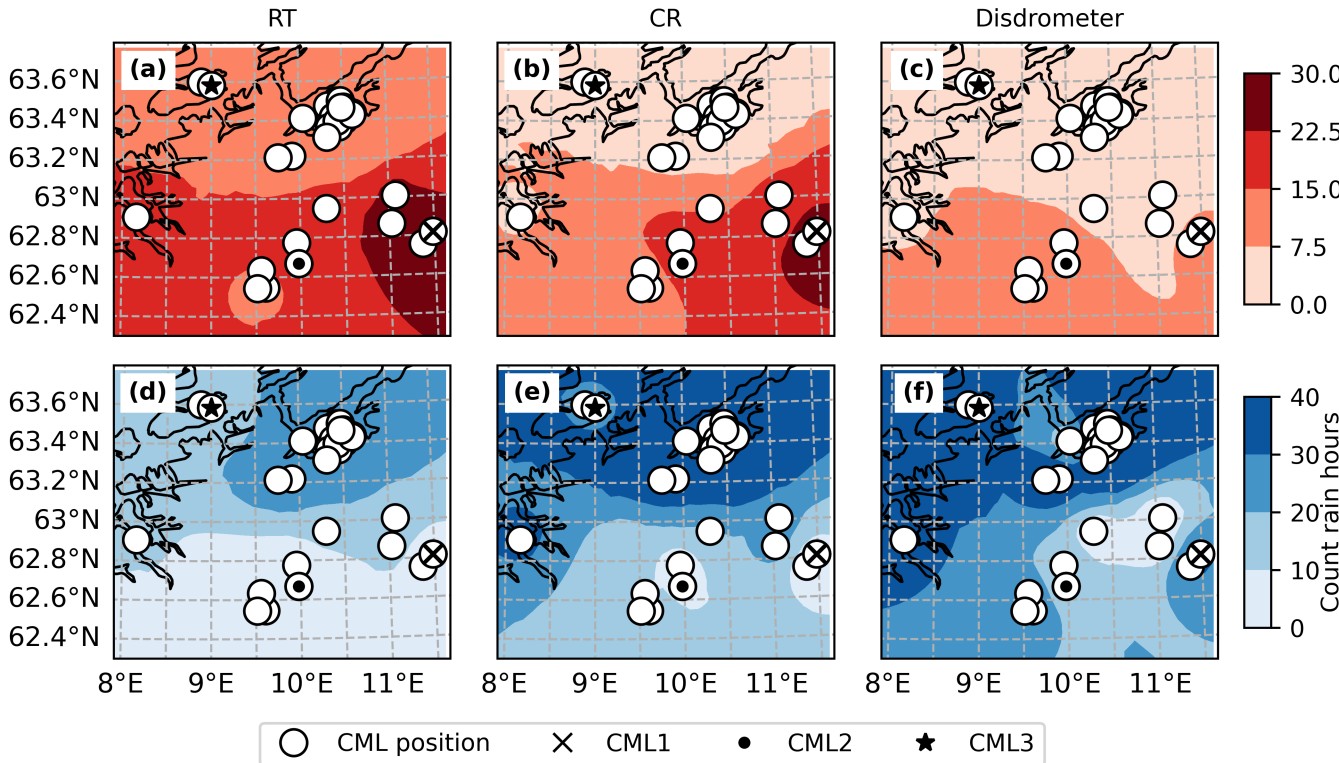

**Figure 2.** Interpolated count of rainy **(a, b, c)** and snowy **(d, e, f)** hours for the RT method **(a, d)**, CR method **(b, e)** and disdrometers **(c, f)** over a 2-day period starting on the 18th of December 2021. The Trondheim fjord and Norwegian coast (black line) are in the north and west, while the Dovre mountains are in the south. White circles indicate the distorted positions (i.e. randomly shifted position to prevent exact retrieval of coordinates) of the CMLs. The black x, dot, and star represent the locations of CML1, CML2, and CML3, respectively, with their time series shown in Fig. 3, Fig. 4, and Fig. 5. The maps were interpolated using inverse distance weighting, with the midpoint of the CML indicating its position.

The difference between the estimated CML rainfall amounts and the radar rainfall amounts is plotted as a function of dew point temperature (Fig. 7 **(a)**). For dew point temperatures below -2 °C, there is a positive bias where the radar generally estimates more precipitation. Between 0 and 2.5 °C, there is a stronger negative bias where the CML estimates more precipitation. For dew point temperatures above 4 °C, the CML and radar estimates show a similar spread. In panels **(b)** and **(c)** the color of the cell indicates the proportion of rainy or snowy hours over total hours for each cell in panel **(a)**. Generally, there is more rainfall in observations above -2.5 °C and more snow in observations below 2.5 °C. Additionally, for events where the CML estimates more rainfall (negative bias), there are more rainy hours as observed by the disdrometer.

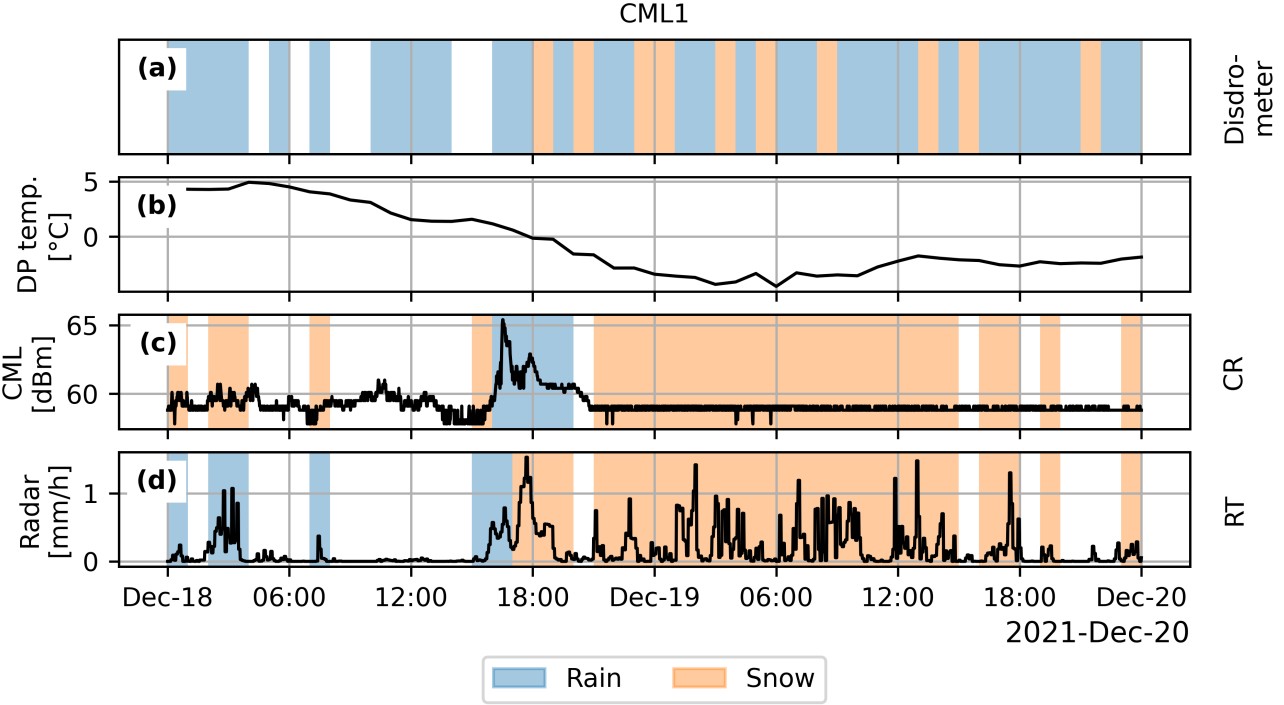

**Figure 3.** Time series for CML1 located in the eastern mountain area of Fig. 2. Dew point temperature (**b**), CML signal loss (**c**), and radar rainfall (**d**). Shaded areas indicate the precipitation type (rain or snow) estimated by the disdrometer (**a**), the CR method (**c**), and the RT method (**d**).

## 3.3 The performance of the CR method vs. the RT method

The performance of each CML-disdrometer pair for the RT and CR methods is compared on the summer and winter datasets using scatter density plots (Fig. 8, (**a, b, c, d, e**)). In the lower row (**f, g, h, i, j**), the mean dew point temperature of each CML disdrometer pair is shown. Note that some cells consist of several CML-disdrometer pairs and that the indicated temperature then is the average of all pairs in the cell. In terms of accuracy (**a, f**), both the CR and RT methods perform similarly well, with a few CMLs performing less well using the CR method. For precision (**b, g**), we observe that, on average, the RT method

outperforms the CR method, indicating that the RT method's positive predictions are more trustworthy than those of the CR method. However, the CMLs where the CR method performs less well tend to have an average temperature above 5 °C, while for colder temperatures, the RT and CR precision scores are more similar. In terms of recall (**c, h**), the CR method performs slightly better than the RT method, indicating that the CR method is better at correctly identifying the disdrometer precipitation type. Looking at the F1 score (**d, i**), which combines the precision and recall score, we observe that the CR method performs

worse on average when the temperature is above 5 °C, but better when the temperature is below 5 °C. Lastly, looking at the MCC score (**e, j**), the CR method outperforms the RT method for most CML-disdrometer pairs.

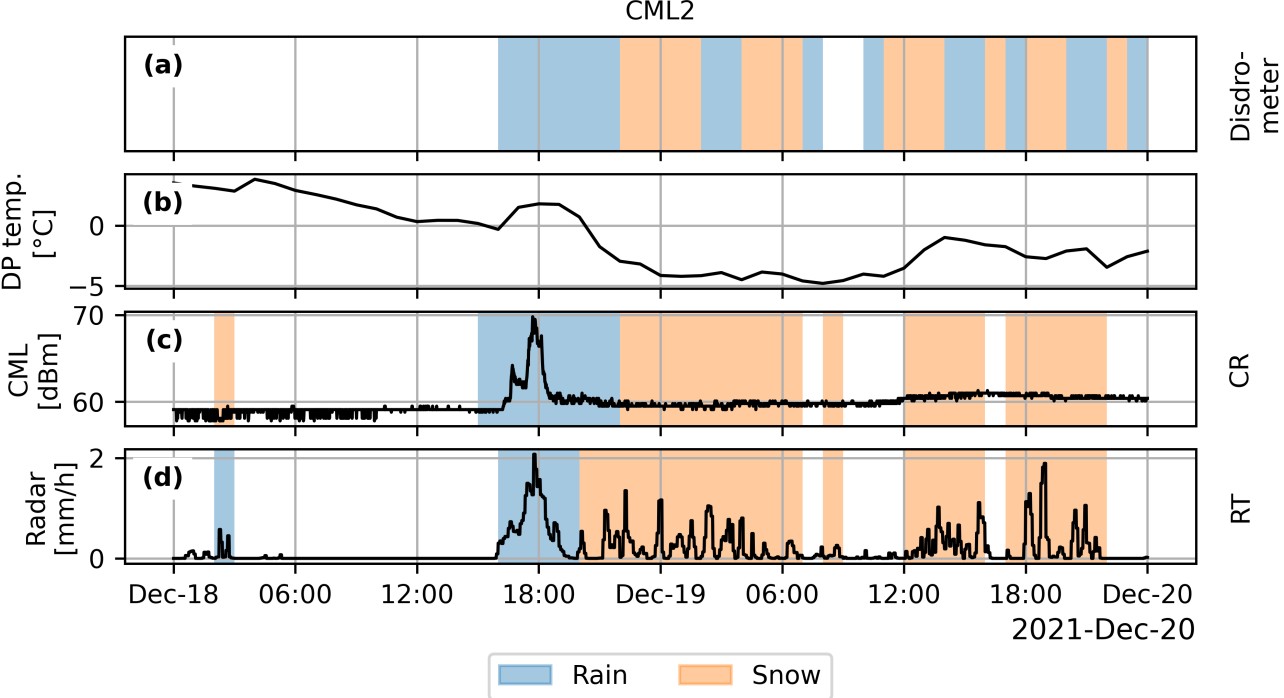

**Figure 4.** Time series for CML2 located in the southern mountain area of Fig. 2. Dew point temperature **(b)**, CML signal loss **(c)**, and radar rainfall **(d)**. Shaded areas in indicate the precipitation type (rain or snow) estimated by the disdrometer **(a)**, the CR method **(c)**, and the RT method **(d)**.

To investigate the effect of temperature on the CR and RT predictions we plotted the multi-label confusion matrix of the two methods for the three temperature intervals -20 °C to -2 °C **(a, d)**, -2 °C to 2 °C **(b, e)** and 2 °C to 20 °C **(c, f)** (Fig. 9). The corresponding accuracy, precision, recall, F1 and MCC scores for the individual classes (rain and snow) as well as the multiclass score are shown in Table 3. We can observe that, compared to the RT method, the CR method generally identifies more correct rainfall events. It also identifies more snowfall events in the temperature interval -2 to 2 °C.

## 4 Discussion

### 4.1 Evaluation of the disdrometers performance

The disdrometer data suggest that rain is more frequent below 0 °C than snow is above 0 °C (Fig. 6), which could indicate that the disdrometer overestimates the number of rainy hours. For example, looking at the CML time series (Fig. 3), the disdrometer shows a mix of rain and snow, while the CR and RT methods indicate only snow. This discrepancy could be due to the spatial distance between disdrometers and CMLs, with temperature differences at these locations possibly causing rain to be recorded at colder temperatures. However, if this effect were significant, we would also expect to see more snow recorded at temperatures

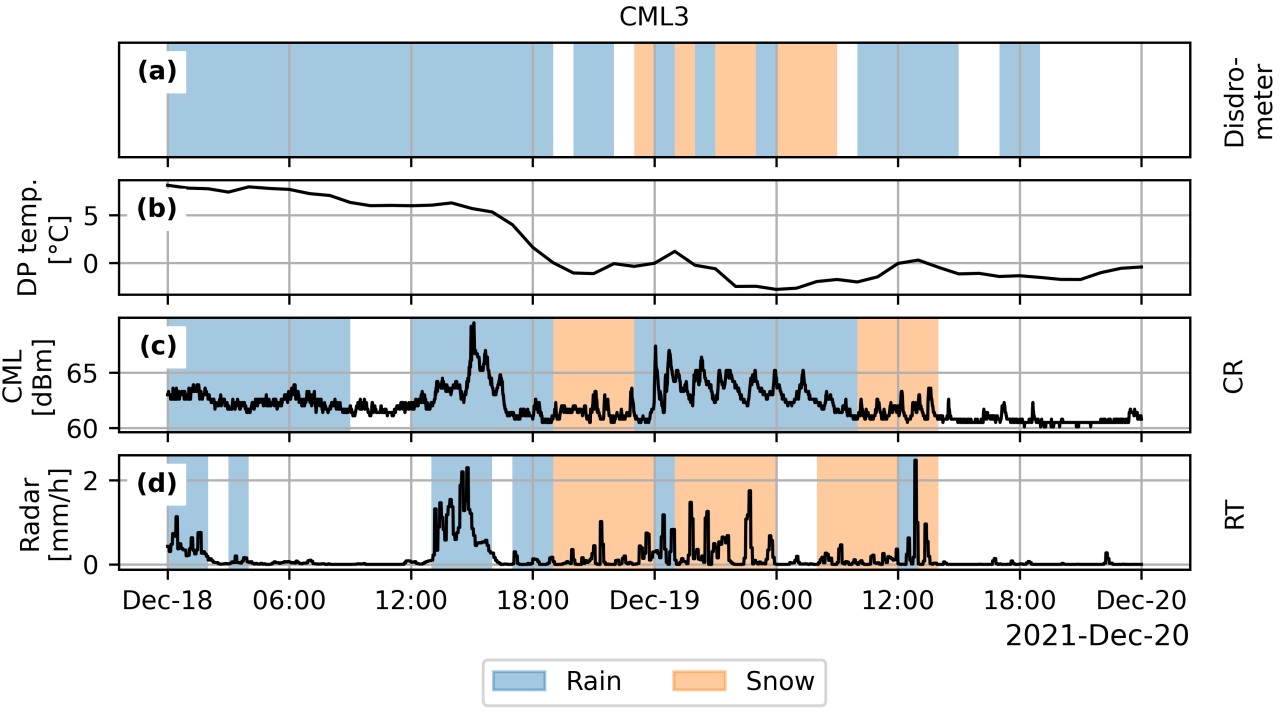

**Figure 5.** Time series for CML3 located in the north western coastal area of Fig. 2. Dew point temperature (**b**), CML signal loss (**c**), and radar rainfall (**d**). Shaded areas indicate the precipitation type (rain or snow) estimated by the disdrometer (**a**), the CR method (**c**), and the RT method (**d**).

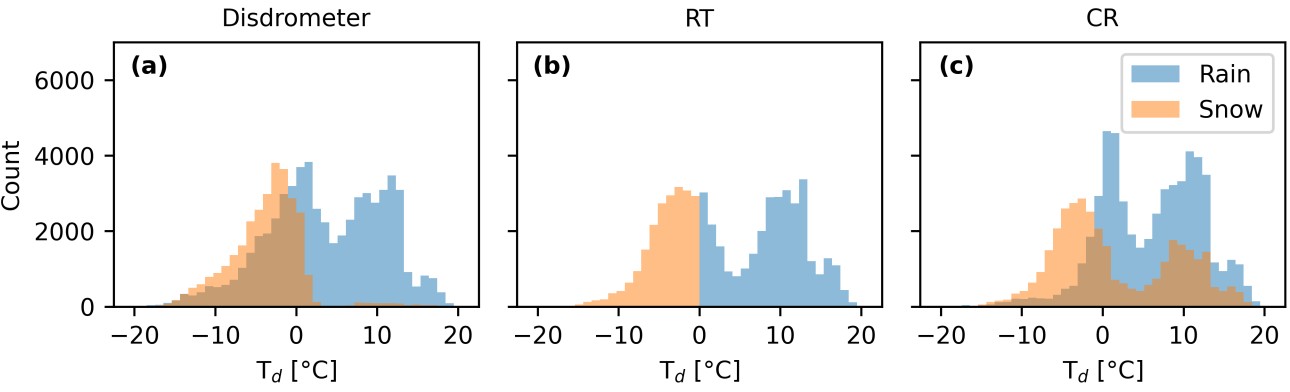

**Figure 6.** Number of hours classified as snowy or rainy for dew point temperature ($T_d$) intervals of 1 °C ranging between -20 to 20 °C for the disdrometers (**a**), RT (**b**) and CR (**c**) estimates.

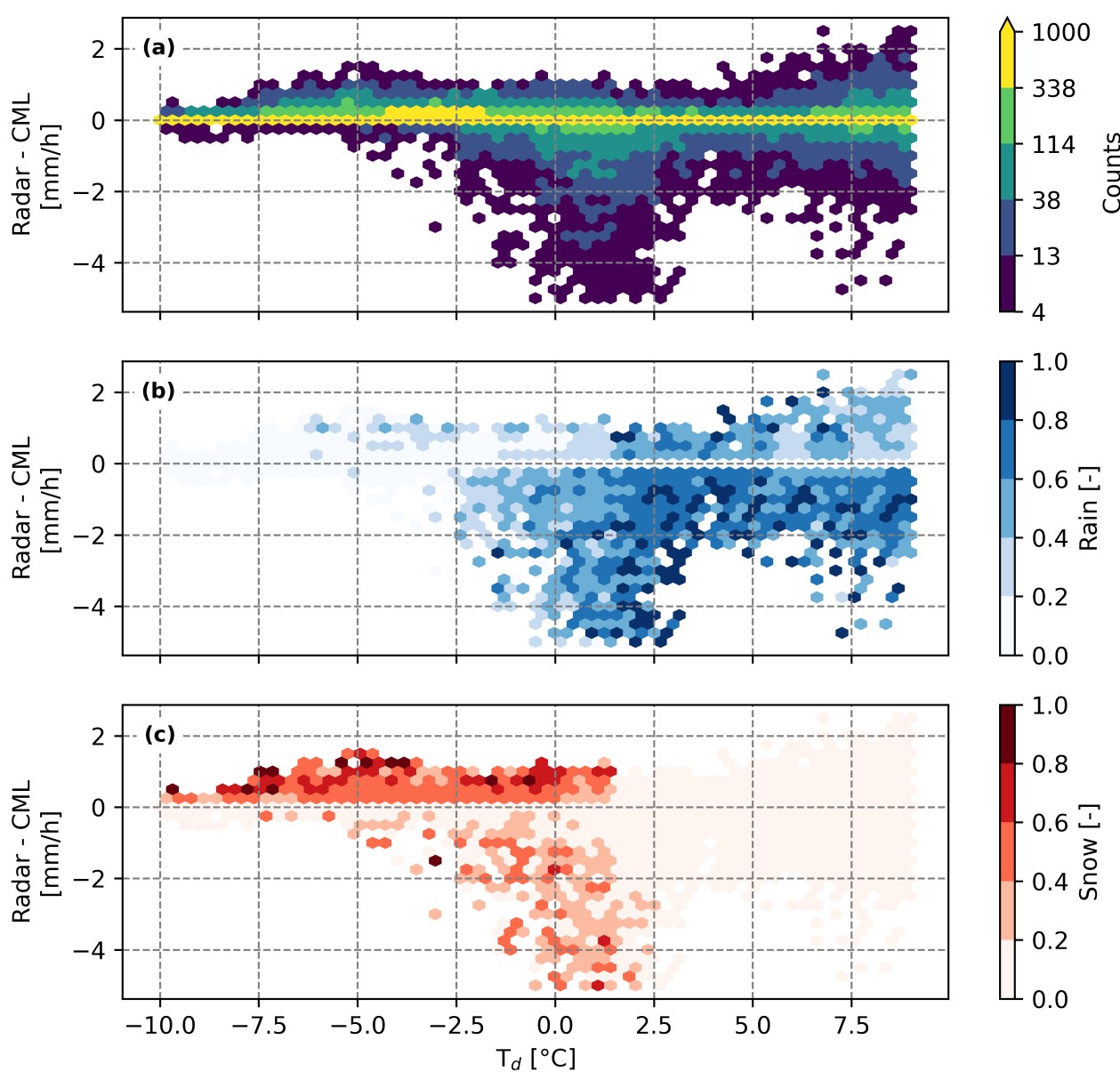

**Figure 7. (a):** Difference between hourly precipitation amounts measured by the radar and CML for dew point temperature ($T_d$) intervals of 1 °C ranging between -10 to 10 °C. **(b, c):** Ratio of rain and snow in each cell as observed by the disdrometer. Cells with less than 4 events are not shown.

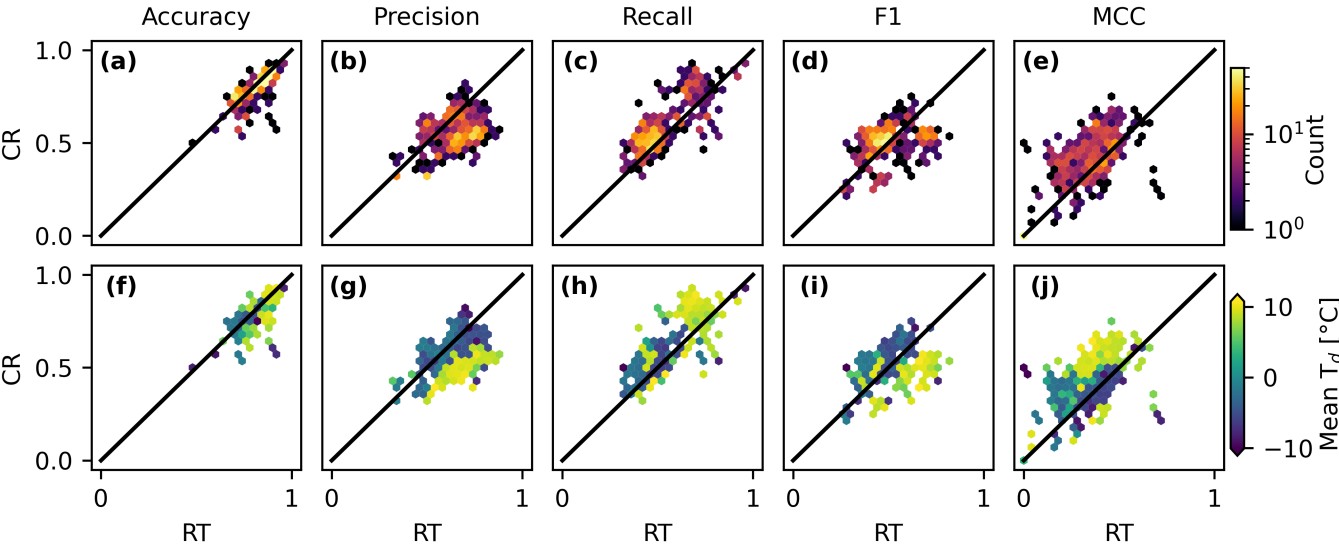

**Figure 8.** Scatter density plots **(a, b, c, d, e)** comparing the accuracy, precision, recall, F1 and MCC score for the CR and RT method for each CML-disdrometer pairs. Average dew point temperature of each cell **(f, g, h, i, j)**.

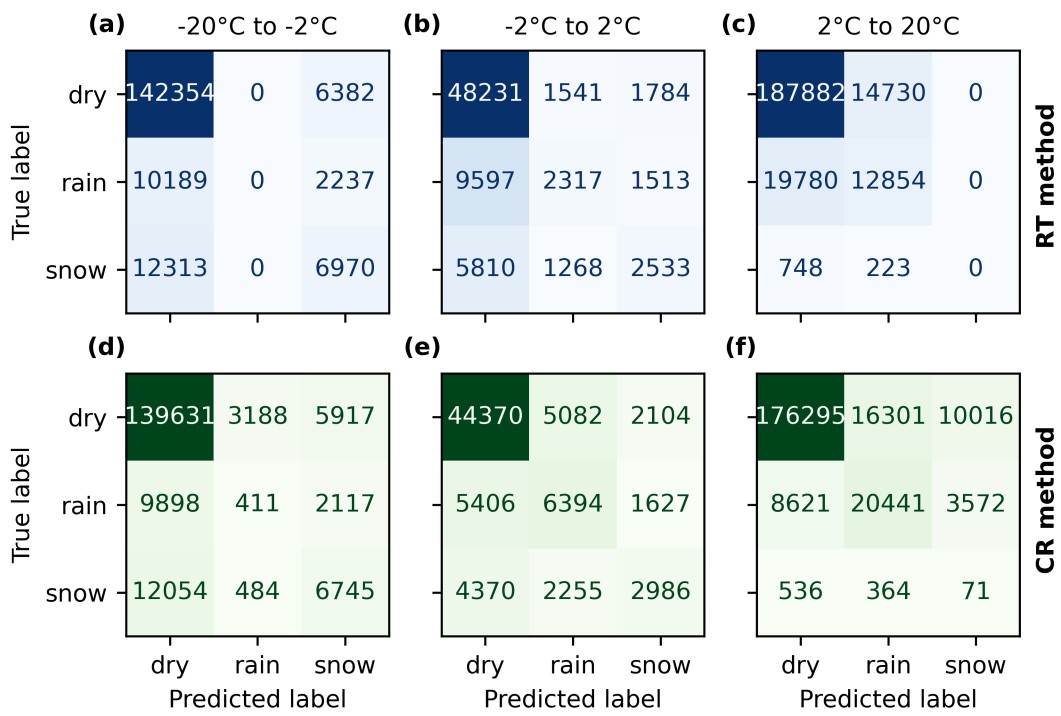

**Figure 9.** Confusion matrices for the RT **(a, b, c)** and CR **(d, e, f)** method for three temperature intervals.

**Table 3.** Accuracy, precision, recall, F1 and MCC score for the RT and CR method for different temperature intervals. The "Rain", "Snow" and "Multi" columns indicate whether the metric was evaluated against rainfall only, snow only or if it used the generalized multiclass metric to evaluate both rain and snow.

| | -20°C to -2°C | | | -2°C to 2°C | | | 2°C to 20°C | | | All temperatures | | |
|---|---|---|---|---|---|---|---|---|---|---|---|---|
| | Rain | Snow | Multi | Rain | Snow | Multi | Rain | Snow | Multi | Rain | Snow | Multi |
| **Accuracy RT** | 0.93 | 0.88 | 0.83 | 0.81 | 0.86 | 0.71 | 0.85 | 1.00 | 0.85 | 0.88 | 0.93 | 0.82 |
| **Accuracy CR** | 0.91 | 0.89 | 0.81 | 0.81 | 0.86 | 0.72 | 0.88 | 0.94 | 0.83 | 0.88 | 0.91 | 0.81 |
| **Precision RT** | 0.93 | 0.69 | 0.66 | 0.65 | 0.67 | 0.55 | 0.68 | 1.00 | 0.68 | 0.68 | 0.70 | 0.59 |
| **Precision CR** | 0.52 | 0.69 | 0.47 | 0.68 | 0.67 | 0.58 | 0.74 | 0.50 | 0.50 | 0.71 | 0.62 | 0.56 |
| **Recall RT** | 0.50 | 0.65 | 0.44 | 0.56 | 0.61 | 0.46 | 0.66 | 0.50 | 0.44 | 0.61 | 0.65 | 0.51 |
| **Recall CR** | 0.51 | 0.65 | 0.44 | 0.68 | 0.63 | 0.55 | 0.77 | 0.51 | 0.52 | 0.70 | 0.64 | 0.56 |
| **F1 RT** | 0.48 | 0.67 | 0.44 | 0.57 | 0.63 | 0.47 | 0.67 | 0.50 | 0.45 | 0.63 | 0.67 | 0.53 |
| **F1 CR** | 0.50 | 0.67 | 0.45 | 0.68 | 0.64 | 0.56 | 0.76 | 0.49 | 0.50 | 0.71 | 0.63 | 0.56 |
| **MCC RT** | 0.00 | 0.34 | 0.30 | 0.19 | 0.27 | 0.28 | 0.34 | 0.00 | 0.34 | 0.28 | 0.34 | 0.32 |
| **MCC CR** | 0.02 | 0.34 | 0.27 | 0.35 | 0.30 | 0.39 | 0.52 | 0.00 | 0.47 | 0.41 | 0.25 | 0.39 |

above 0 °C, which we do not observe. Another explanation for these discrepancies could be the aggregation method, where rainy hours are given priority over snowy hours, creating the impression that there are more rainy hours below 0 °C. However, if the disdrometer was correct and 10 minutes of an hour were indeed rainy, that would still represent a significant amount of rain during that hour. Moreover, comparing the CR rain distribution with the disdrometer rain distribution (Fig. 6), we can see that the CR method observes fewer rainy hours below 0 °C, indicating that the disdrometers overestimate the number of rainy hours below 0 °C.

One explanation for this overestimation could be that wet snow makes the disdrometer alternate between snow and rain, creating the impression that there is more rain below 0 °C. Moreover, different precipitation type sensors are also known to disagree during mixed precipitation events (Bloemink, 2005; Pickering et al., 2021), indicating that these events are hard to classify. However, while wet snow could explain some of the rainfall events below 0 °C, the large proportion of rainy hours below -10 °C remains puzzling, as wet snow is not expected at such low temperatures. Another explanation could be misclassification by the disdrometer due to factors such as strong winds, particles falling through the edges of the sampling area and splashing. Although correction algorithms exist (Friedrich et al., 2013), they typically require the full velocity-drop size distribution matrix, which our disdrometers do not provide. This limitation may lead to less accurate classifications. Additionally, cars spraying water from salted roads could contribute to the high number of rainfall events recorded below 0 °C. Consequently, while the disdrometers provide valuable estimates, they do not perfectly represent the ground truth, especially during mixed precipitation events.

## 4.2 The case study

The case study indicated that the CR and RT methods estimated a similar number of snow events in the mountains, while along the coast (for instance at CML3), the CR estimates were more in line with the disdrometer estimates, suggesting an advantage of using the CR method in certain climatic zones. This could be due to the warmer conditions along the coast, which keep the temperature around 0 °C, a range where the RT method has more uncertainties. However, looking at the time series of CML3 (Fig. 5) we see that the CR method estimates rainfall during the true snow event, lasting from 00:00 to 10:00 on the 19th of December. It also wrongly estimates snowfall before and after the true snowfall event. Thus, while the case study map suggests a better agreement between the CR and disdrometer estimates (Fig. 2), there is still a significant discrepancy between the CR and disdrometer estimates in the hourly time series (Fig. 5). This discrepancy could be due to the spatial difference between the CML and disdrometer or disdrometer misclassification, as discussed above. Another explanation could be that the disdrometers classify mixed precipitation, such as wet snow, as snow, while the CMLs classify wet snow as rainfall, leading to a misclassification by the CMLs. This phenomenon is clearly observable during the true snowfall event lasting from 00:00 to 10:00 on the 19th of December (Fig. 5), where the CML estimates a long rainy period and the disdrometer estimates a mixture of rainfall and snow.

## 4.3 The CR and RT methods classification performance

For snowfall classification at temperatures between -2 and 2 °C, the CR method holds a slight advantage over the RT method, as reflected in a slightly higher MCC (0.3 vs 0.27) and recall score (0.63 vs 0.61) (Table 3). This indicates that the CR method is slightly better at identifying true snow events (higher recall) while maintaining reliable estimates (equal precision of 0.67). Looking at the confusion matrix for the same temperature interval, we observe that compared to the RT method, the CR method correctly identified 2986 true snowy hours, an increase of 453 hours, while wrongly classifying 3731 hours as snow, an increase of 434 hours (Fig. 9). In other words, the CR method identifies more true snowfall events while also wrongly classifying rainy and dry events as snow. The large number of false snow events estimated by the CR method, also observable above 2 °C (Fig. 6), might be due to several factors. Low-intensity rainfall events could fail in triggering the CML rainfall detection algorithm, for instance due to the quantization of the CML signal. Further, due to the spatial difference between the radar beam and the CMLs, the precipitation might hit the radar, but miss the CML, triggering the CR method to estimate snow. Finally, hardware issues with the CML, or database errors, could result in a flat signal level, causing the CR method to misinterpret conditions and estimate snow. Better quality control of the CMLs, for instance by checking their correlation against the weather radar during rainfall events could improve the CR estimates. Next, at temperatures below -2 °C (Fig. 6), both the CR and RT method snow classification produced recall scores of 0.65, indicating that the weather radar misses many of the snowfall events recorded by the disdrometers. This could be due to the spatial difference between the CML and disdrometer or disdrometer misclassification, as discussed above. Another explanation could be that blowing winds or road traffic transport snow horizontally, causing the disdrometers to detect snow that does not originate from the sky.

In terms of rainfall classification, the CR method performs as well as or outperforms the RT method for all temperatures above -2 °C (Table 3). For instance, for temperatures above 2 °C, the binary accuracy score for rainfall increases from 0.85 with the RT method to 0.88 with the CR method, and the binary MCC for rainfall increases from 0.34 to 0.52. However, both the RT and CR methods still miss many of the rainfall events observed by the disdrometer, which is evident in their recall scores of 0.66 and 0.77, respectively. As discussed above, this could be due to the spatial difference between the CML and disdrometer, or for instance splashing from the roads, leading the disdrometers to estimate rainfall that is not detected by the CML or weather radar. The increased performance of the CR method, compared to the RT method, could be due to radar overshooting or the radar beam being blocked by mountains.

For classifying snowfall and rainfall, both methods have their own strengths and weaknesses. The CR method shows a better ability to classify precipitation in the interval -2 to 2 °C (MCC = 0.39), but falsely estimates a large number of snowfall events above 2 °C. The RT method, on the other hand, provides reliable precipitation classification below -2 °C (MCC = 0.30) and above 2 °C (MCC = 0.34), while its performance is not as good within the interval -2 to 2 °C (MCC = 0.28). Consequently, combining the RT and CR methods would be optimal. This could, for instance, be done by using the CR method in the interval -2 to 2 °C and the RT method below -2 °C. Above 2 °C, precipitation could be classified as rainfall if either the RT or CR method detects rainfall.

## 4.4 Uncertainties, the impact of the aggregation method and mixed precipitation types

The impact of mixed precipitation, such as wet snow, on the observation methods (disdrometers, CR method, and RT method) remains a significant source of uncertainty in this study. This uncertainty arises from the fact that none of the observation methods can reliably classify mixed precipitation. Additionally, accurately classifying hours that may contain both snow and rain when aggregating the data to hourly intervals poses a challenge. One solution is to introduce a mixed class, classifying hours with both snow and rain as mixed precipitation. However, it remains unclear whether true mixed precipitation, like wet snow, would consistently cause the disdrometer, CR, and RT methods to alternate between detecting rain and snow, which could make the mixed class less physically meaningful and lead to inconsistent representations across different estimation methods. While other studies, such as Pickering et al. (2021), aggregated multiple precipitation type data from different sensors to longer periods using a boolean algorithm, we found that the sensors used in our study differed too much for a similar approach. For instance, the RT method uses temperature data with hourly resolution, which complicates accurately capturing hours with both wet and solid precipitation types. While some studies introduce mixed precipitation estimates by classifying precipitation within a fixed temperature interval as mixed (Harpold et al., 2017), the true precipitation type within this interval could still be purely rain or purely snow, leading to inaccurate classifications. Furthermore, since the radar might estimate precipitation slightly before the CML, the CR method is prone to estimate snow before rainfall events, leading to an overestimation of mixed precipitation. This could be addressed by aggregating the CML wet period so that the radar precipitation estimates fall inside the wet period, but this would require further tuning to avoid estimating too many rainfall events at the expense of fewer true mixed events. This work uses a simplified aggregation method in order to avoid introducing too many parameters.

We found that, while this approach produces a higher rain-to-snow ratio for negative temperatures, the assumptions are stated more explicitly, and the final results and conclusion remain similar to what we got using other aggregation methods.

Around 0 °C, the CMLs estimate larger rainfall amounts compared to the weather radar (Fig. 7). The same temperature interval is also characterized by the disdrometers observing both rainfall and snow. This suggests that significant discrepancies between the CML and radar estimates around 0 °C may be attributed to wet snow. This effect has been observed before in previous studies, such as Overeem et al. (2016) and Graf et al. (2020), where a marked positive CML bias during the winter months was observed. Further, Fig. 7 reveals that the CML bias as a function of temperature, follows a smooth transition. This indicates that wet snow, as observed by the CML, does not belong to a homogeneous group, but instead follows a gradual transition from snow to rainfall while melting.

Another source of uncertainty lies in the temperature data used for the RT method. The temperature data is a downscaled version of ERA5 data that is combined with ground observations. Lussana et al. (2019) found that the expected RMSE of the temperature data ranged between 1-2 °C in observation dense areas and 2-2.5 °C in observation sparse regions. The RT method performance could thus be less good in areas with complex terrain and sparse ground observations.

While studies such as Gjertsen and Ødegaard (2005), Casellas et al. (2021), and Saltikoff et al. (2015) have evaluated the performance of temperature-based precipitation phase classification methods, these studies typically vary in methodology, terrain complexity, radar technology, and the instruments used to estimate the ground truth. This variability introduces challenges when comparing results across different studies. Although the disdrometers used in this study provide a large dataset, they have some limitations; in particular, the large number of rainy events recorded below 0 °C and the lack of mixed class classification introduce uncertainties specific to this study. Further, any type of wet precipitation can cause the CML signal level to drop, potentially leading to precipitation being falsely classified as rainfall. Another source of uncertainty is the spatial distance between the disdrometer and the CML, where for instance temperature variations due to elevation and spatial differences can affect precipitation classification. Moreover, the temperature model used is based on model data and could be improved by using ground-based sensors. Combined, these factors introduce large uncertainties in this study, and further make it challenging to directly compare the CR and RT estimates in this study to those from similar studies.

Nevertheless, our study demonstrates that CMLs can be used to enhance the classification of snow and rainfall around 0 °C, which are useful for hydrological applications such as in predicting hydropower production, flooding, avalanches and slush avalanches.

## 5 Conclusions

In this work, we have compared two methods for classifying rain and snow. The "radar-temperature" (RT) method works by classifying weather radar precipitation below 0 °C as snow and above as rain, using dew-point temperature derived from downscaled ERA5 data. The "CML-radar" (CR) method, exploits the fact that dry snow causes minimal signal attenuation in the CML signal level, and works by classifying time steps where the weather radar detects precipitation, and the CMLs do

not detect precipitation, as snowy. Time steps where the CML detects rainfall are set to rainy. The estimates were compared to estimates from nearby disdrometers located along roads in Norway.

Our results show that the CR method outperforms the RT method for dry snow detection between -2 and 2 °C and, in general, for rainfall detection, suggesting that CMLs can be used to better classify rain and snow. Further, our results indicate that wet snow is classified as rainfall by the CMLs and that during these events the disdrometers tend to estimate a mix of rainfall and snow. Future work should investigate methods for CML wet snow detection, preferably using several different precipitation-type sensors as ground truth, as suggested by Pickering et al. (2021). Future work should also investigate how the CR estimates

impact hydrological models.

Overall, our findings suggest a new application for using CMLs to identify dry snow and contribute to a better understanding of how CMLs behave during events of mixed precipitation.

*Code and data availability.* The software used for CML processing software is available under https://github.com/pycomlink/pycomlink/tree/master. Disdrometer data is available from (Frost, 2024). Radar data is available from (THREDDS, 2024). CML data were provided by

Ericsson and are not publicly available.

*Author contributions.* Conceptualization: EØ, JA, RB. Data curation: EØ. Methodology: EØ, JA, RG, RB, MW, NOK, CC, VN. Software: EØ, CC. Supervision: VN, MW, NOK, RB. Writing – original draft preparation: EØ. Writing – review and editing: EØ, RG, JA, RB, MW, NOK, CC, VN.

*Competing interests.* The contact author has declared that neither of the authors has any competing interests.

*Acknowledgements.* The authors thank co-supervisor Etienne Leblois for nice discussions. We would also like to thank Ericsson for providing CML data. This work is funded by the Norwegian University of Life Sciences and the German Research Foundation via the SpraiLINK project (Grant CH-1785/2-1).

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
