# Peer review of "Combining commercial microwave links and weather radar for classification of dry snow and rainfall"

_EGUsphere, 2024_

## Referee Comment (RC1)

Review of: "Combining commercial microwave links and weather radar for classification of dry snow and rainfall" by Erlend Øydvin et al. (2024)

This paper presents an interesting novel method to classify dry snow and rainfall by merging two precipitation measurement techniques. Previous studies has shown the difficulty to estimate precipitation phase, yet is a relatively important process in the hydrological cycle. The authors perform an elaborate analysis of their novel method and compare this with two more established methods. In general, this is a well-written manuscript. However, some parts would benefit from a revision, in order to improve the overall quality of the manuscript. Below I have stated more general and specific comments, which I hope the authors consider to be constructive.

**General:**

- Results:
  Parts of the result section are hard to understand due to the combination of complex figures in combination with it being unclear to which subplot is being referred. It would already be helpful if all subplots are labelled with a,b,c… (which might be a requirement by AMT, check their website) in combination with adding in-text references to the specific subplots. Moreover, for the reader, especially those who are less familiar with CMLs, it would be perhaps worth considering to add a basic timeseries (like Fig. 6) to the start of the results section, in order to show the reader how the methods workout in reality and help get an idea about the differences between the CR and RT method. In fact, such a figure already illustrates a lot and prepares the reader for the following more in-depth analysis. Additionally, for some figures it is not clear how you used the data to create the figure (e.g., Fig. 2, see specific comments).

- Road surface conditions:
  I am struggling to see the added value of Sect. 3.5 in which the road friction is related to the classification methods (except the discussion of Fig. 6 which helps understanding the methods). I understand that the new method could also be used to estimate road conditions and I do agree that it is a nice illustration of the potential of these methods, but I think discussing road friction would require a more elaborate analysis. As you write in the discussion, road conditions are not simply a result of dry snow or rainfall, but also previous conditions (e.g., was there already a snow pack, has there been any snow/ice removal, road temperature preceding rain event) while also other precipitation phases should be considered (e.g., what about freezing rain?). Also until Sect 2.6, there hasn't been any reference of the road conditions, so it came as a bit of a surprise. Based on the introduction, I would have expected that your new method would have been applied to something like a hydrological model.

- Discussion:
  I like that the discussion is relatively concise, but it does not include many references to previous studies on similar topics. I encourage you to include some references to previous studies, so that the reader would be better able to place your results into a wider perspective. For example, how do your findings compare with other studies that use similar methods (mostly for the RT method)? See also the specific comments.

**Specific comments:**

- L9: perhaps include the location of your study.
- L12: There is no mention of the application to road conditions.
- L54-58: You introduce the dewpoint temperature here, but would it be an idea to include the explanation as to why both are important (L19-33) here? This will help the reader relate the dewpoint temperature to the underlying processes. Or perhaps even referring back to the previously mentioned importance of profiles would already help the reader.
- L73-85: I suggest to include some more studies where they show the influence of various precipitation types on the CML signal intensity in more detail.
  See for example:
  Hansryd, J., Li, Y., Chen, J., & Ligander, P. (2010). Long term path attenuation measurement of the 71–76 GHz band in a 70/80 GHz microwave link. *Proceedings of the Fourth European Conference on Antennas and Propagation*
  van Leth, T. C., Overeem, A., Leijnse, H., & Uijlenhoet, R. (2018). A measurement campaign to assess sources of error in microwave link rainfall estimation. *Atmospheric Measurement Techniques*, *11*(8), 4645-4669. https://doi.org/10.5194/amt-11-4645-2018
- L79-80: Based on this line, I would have expected the paper to focus on applying/elaborating on the methods of Cherkassky and Ostrometzky. I suggest to rephrase this.
- L83: reports → reported, in order to use the same tense as the previous references (very minor comment)
- L90: I would suggest to already mention here that you cannot share the location of the CMLs due to data security reasons.
- L103-104 (and L118-120): I struggle to understand why you do this. As I understand you extend the wet periods in order to make sure that the rain event has also completely passed the disdrometer, is that correct? Yet in your analysis, there is no comparison of rain events but you compare individual time steps and average/summed periods, so I don't see how this period extension exactly works. Or am I missing something here?
- L107: Could you explain briefly on what concept the method of Leijnse et al. (2008) is based? Not every reader will know how their method works.
- L110-114: Could you provide a reference to this radar product for readers who would be interested to know more about this product?
- L113-115: Has the seaclutter and other large peaks been removed by you or was that already done in the data you downloaded?
- L121-136: How well do these disdrometers work? Are there any known biases or uncertainties? Especially because in Sect. 4.2 you refer to their potential uncertainty. This could also be discussed in the discussion.
- L132: Why do you use 8km? Is there a specific reason for this? Is this based on previous studies, or based on your own data? Or is this a common value in CML studies?
- L138: Are there any common biases or uncertainties in this data? If so, please mention to help the reader.
- L147: Marks et al. use 0 C, but that is for the USA right? Why would that be applicable to Norway? As you write in your introduction, these threshold values can have a relatively large range, and thus can have a relatively large influence on your results.
- L155: Perhaps show the equation for MCC. I think not all readers will be familiar with it.
- Fig 2.: The caption is hard to understand here, because you refer to RT and the disdrometers both as references. Also, it is not clear what you mean with "the MCC's are computed for

each CML-disdrometer pairs using 1 month of data". What do you mean with 1 month of data? Just the December data? It almost seems like there is an additional step between the data described in Sect. 2 and this figure. (also minor comment: I think it should be pair instead of pairs)

- Fig. 3: Is this the average of the MCC's for all CML-disdrometer pairs as a function of Td?
- L198-210: Here the addition of in-text references to subplots a,b,etc. would help to guide the reader.
- L211-221: I got confused here because of the text in the parentheses. Why are both the RT and the CR method referred to as CML rain/snow/dry? Wouldn't it be an idea to just refer to them as rain/snow/dry? Or is there a specific reason you included CML?
- L214: to be fully correct the radar observes precipitation instead of rain (very minor comment)
- Fig 5. I understand that you cannot share the exact location of the CMLs because of data security reasons, but is there the possibility to describe the landscape a bit? For example, are there any large elevation differences (i.e., mountains/fjords) in this area that could for example create orographic precipitation or cause beam blockage of the radar? Additionally, I suggest to add titles to the columns, so that it is immediately clear which method is shown in which column.
- Discussion: Are there any previous studies in which the RT method (or something similar) has been compared with disdrometers (or other precipitation phase observations). If so, I would recommend to include a short discussion on this. How do your results compare to those studies? Are they similar or is it the RT method more difficult in Norway because of for example the elevation differences? Such a discussion would allow to put the performance of the CR method into a wider context.
- L264: the circle of degree Celsius should be in superscript (very minor comment)
- L266: How often does a disdrometer misclassify precipitation phase? I would advise you to include a brief discussion based on previous literature regarding uncertainty when using disdrometers to estimate precipitation phase.
- Fig 6: In the bottom subplot, I suggest to use a different shading color for the diff<-5 mm, because it overlaps with the shadings above. Or perhaps you can even leave this out, as you do not discuss this in the text if I'm correct.
- L270-285: Based on this section, it seems that only wet snow could cause misclassifications. I would suggest to make clear that any form of wet precipitation causes the CML signal to drop. Additionally could it also be that spatial temperature and humidity differences can cause that at the disdrometer location dry snow is falling while somewhere else the precipitation has started melting? I can imagine that this happens when temperatures are in the transition region between rain and snow.
- L288-289: Are there any previous studies that show differences between radar and CML/disdrometer measurements? If so, I suggest to discuss these here.

---

## Referee Comment (RC2)

**Review of the following manuscript**

Title: Combining commercial microwave links and weather radar for classification of dry snow and rainfall
Author(s): Erlend Øydvin et al.
MS No.: egusphere-2024-2625
MS type: Research article

This manuscript proposes simple methods to classify the precipitation type (i.e. rain/snow/dry) based on opportunistic data collected by wireless microwave links in combination with radar data (what the authors called "CR method"), relying on the different sensitivity of microwave links and weather radars to snow.

Gathering valuable information about snow from CML data has not been really addressed so far. Hence, the topic of this manuscript is interesting. Using joint radar and CML data for meteorological purposes is interesting as well. So, I think there is enough novelty and interest in this contribution.
However, I see a number of points that should be addressed by the authors before publication. Therefore, I recommend to accept the manuscript after a major revision.

**Major points**

1) Datasets. The authors put together data from 2179 CMLs across Norway, later reduced to 550 and 435 for the winter and summer datasets (Sec. 2.3). However, they didn't provide much information about the characteristics of these links (they just mentioned lower and upper bounds on lines 135-136 of their manuscript). In particular, it would be good to provide the frequency vs length distribution and the quantization error, which determine the minimum detectable rainfall intensity as well the accuracy of CML measurements. I know that CML metadata have some issues due as the owners are private companies as Ericsson, but I saw these data published in several papers on this subject. Hence, we do not know which is the sensitivity of these links to light precipitation rates, which are typical of the winter period where also snow is present in cold regions. This would be also important to understand to what extent the scatter in the data highlighted for instance in Figs. 2-and 4 is due to some/several/many CMLs performing worse than others as their sensitivity is lower, or whether it is due to the classification method itself.

2) Methods. In particular, the way labels (i.e. precipitation type) were assigned. If I got it correctly, CML data are sampled every 1-min, radar data are available every 5-min, disdrometer data are provided every 10-min and meteorological data come every 1 hour. Due to the different sampling rate of the sensors involved, it is necessary to define a suitable time window within which labels are assigned.

   a) I would like that the authors clearly state how they put together the 1-min CML slots over the integration window (arguably 1-hour) to decide whether an hour is rain or anything else. The same should be done for the other sensors. I think these concepts are written in Sec. 3.4, but it's not the right place in my view. This is about methods. Moreover, the author should provide some justification for all those threshold values they used. Another important detail on methods is somewhat hidden as it is delayed to lines 290-292.
   b) Related to previous point a): it is not easy (at least to me) to understand how the authors assigned the precipitation type labels to get the results in Fig. 2 just based on what the authors wrote in previous Sec. 2.
   c) Not clear which is the sampling rate of disdrometer data. On line 128 it is written that "The disdrometers […] provide an estimate of the precipitation type every 10 minutes". However in Fig. 1 it seems that data are every 1-min. Please clarify.

3) Results: I think the results in Sec. 3 are not presented in the best way. Reading the abstract, the purpose of this paper is clear: "This study introduces a new approach to improve rainfall and dry snow

classification by combining weather radar precipitation detection with CML signal attenuation […]. Both methods were evaluated using ground measurements from disdrometers.". Hence, I expect a basic performance assessment of the RT and CR classification methods proposed by the authors against the ground truth. However, looking at Figs. 1-6, well it's not very clear to me.

    a) I agree MCC is a comprehensive performance indicator but it's not easy to understand how MCC values in Fig. 2 turn into good or bad labelling of data. I think a simple contingency table with indexes as Specificity and Recall for either method would help.

    b) Fig. 3 puts together hundreds of CMLs (which I guess have different performance as rainfall sensors). The trends in the figure can be identified looking at Fig. 2. I don't think it brings a lot of extra information. Indeed there are just a couple of short statements in the text that comment this figure.

    c) Fig. 4: I didn't get well what is shown in rows 2-4 of this figure from the explanation in the text. On lines 205-207 it is written that "In the second, third and fourth row, we have plotted the fraction of hours within the bins where the disdrometer recorded at least 10 minutes of rain, snow and both snow and rain (mix) respectively." The term "bins" is maybe inappropriate. I would use it for an histogram. Let me see if I got it. First, the counts in the top row are now hours rather than minutes. Right? Then in rows 2-4, these counts have different colours according to the fraction of time hours were flagged as rain/snow/mix by the disdrometer. So, if an hexagon in the first row has a color corresponding to 10 counts, that is 10 hours, the same point in the second row is colored according to the fraction of this 10 hours flagged as rain by the disdrometer? I don't know if I got it. Whatever is the case, please explain it as I tried to do, as it is hard to understand it from the short explanation in the text. Moreover, the colorbar is not the best. I would have used a blue scale for fractions < 0.5 and a red scale for values > 0. Having stated this, the results in Fig. 4, at least what one can notice first sight, is expected in my view: a lot of rainy time above 0°C and a lot of snowy time below 0°C and finally an uncertainty region around 0°C. Moreover, we cannot say whether a 2.5 mm/h difference in CML vs radar accumulation is large or not if we do not know the exact magnitude of rain accumulation. Looking at some finer features:  it is a bit strange that radar overestimates so much in several cases at large positive Td values. Is it maybe that those were high precipitation summer events? Hence, the fractional difference Radar-CML is much less.

    d) Fig. 5: I see really little information here. What's the purpose of showing this figure? On the other hand, I think Fig. 6 is very useful as it shows the whole story as seen by the different sensors. I would have moved this one forward as Fig. 2 because it is very easy to understand and helps the reader in interpreting better the scatterplots. Good job here!

4) Discussion.  More than issues, here I am just pointing some other possible explanations of the results.

    a) Sec. 4.1. The authors argue that disdrometer could fail as ground truth in some cases. In addition, should we 100% trust estimates of Td based on ERA5? These are not ground data measured by weather stations. Maybe it would be good to check ERA5 RH and Ta outcomes against some weather stations that are for sure available (for instance as you did in the third row of Fig. 6). Indeed, looking at Eqn. (1), Td can range between -2 and 0°C with Ta=0°C and RH ranging from 90 and 100%. It means that a 10% error on RH measurement/estimate turns into a 2°C error in Td estimate.

    b) Sec. 4.2 (but also Sec. 4.1). The only way to check the effect of spatial distance between CMLs and disdrometers is to select a subset of CMLs with a disdrometer in the neighbourhood and calculate the MCC:

    c) About wet snow. Looking at Fig. 4 wet snow seems to correlate with Td (as expected). I expect wet snow to occur at small negative values of Td, while dry snow to occur at colder temperatures. Below -5°C it's mostly dry snow according to Fig. 4. Can the author refine a little bit their decision algorithm including a Td threshold to discriminate between wet and dry snow?

    d) Could it be that some unexpected results are due to the way data with different sampling rates were combined together in the hourly time windows and the threshold values used to flag rain/snow /mix/dry intervals)

**Minor comments**:
Line 44: "Human observations can be subjective and aren't suitable for continuous high frequency monitoring"
The term "high frequency" is a bit ambiguous in this context. I guess you mean high-rate monitoring.

Lines 174-175: I cannot understand the statement. "Our dataset consists of CML-disdrometer pairs from the summer dataset and the winter dataset. Every minute each pair provides several different observations such as disdrometer observed precipitation type, dew point temperature and CML signal loss.". Isn't the dew point temperature taken from the ERA5 dataset? "Our dataset" in my eyes are the data "we produced ourselves", but it's not the case. Why radar maps are not part of "our dataset"?

Line 174: you state that disdrometer data are taken every 1-min while previously (line 128) you stated that precipitation type classification from disdrometers is available every 10-min. Please clarify.

Lines 179-182, comment on second row of the figure. I would say "snow mostly below 0°C" while "rainfall is mainly above 0°C".

Line 186: "using monthly time series". To me "monthly time series" means that you have used a time series long one month and extract information from the time series as a whole. Please explain-

Line 187: pair instead of pairs

Figure 2: the gray cluster is hardly visible in the two bottom panels. It would be better to use a darker tone of gray.

Lines 186-192 (Comments of Figure 2): I see MCC_rain of CR is usually better when it's above 0.4, maybe worth to state it-.

Lines 193-196(Comments of Figure 2): the first thing I noticed looking at this figure is that CML really enhance MCC_rain wrt radar (as expected).

Figure 5, caption: I guess it is "mix(red)" instead of "dry(red)".

Line 264: 10°C instead

Line 453: DOI is not correct

---

## Author Comment (AC1)

The authors would first of all like to thank the reviewers for their thoughtful and constructive review. We note that the goal of our manuscript was to evaluate the performance of a novel method for classifying precipitation types, that works by combining estimates from weather radar and commercial microwave links (CR method), and compare it to ground truth disdrometer estimates located along roads in Norway. This method was compared to estimates from weather radar and a downscaled ERA5 temperature model (RT method) against the same disdrometers. Since the different data sources have different temporal resolutions and spatial representations, they must be aggregated in order to be comparable. Our main take away from the review was that this aggregation should be performed in a clearer and more understandable manner, and that parameters used in the CR method (length of wet period) and RT method (temperature threshold) should be better justified.

In the original manuscript the observation methods (CR method, RT method and disdrometer ground truth) were compared using 1 minute resolution and the observation methods estimates were resampled so that they could be compared at that scale. While we think that this is a valid approach, we acknowledge that it would be easier for the reader to understand the aggregation method if the observation methods were compared at 1 hour resolution, as suggested by one of the reviewers. One aspect that arises from aggregating the disdrometer data to hourly resolution is how to classify hours where there have been observed a mixture of rain and snow. An intuitive approach would be to classify these hours as mixed precipitation. However, none of the observation methods used in this experiment directly estimates true mixed precipitation, such as wet snow, and it is therefore not known how these true mixed events show up across the different observation methods. Introducing a mixed class thus requires tuning of each of the observation methods. We have experimented with this mixed class, but found that since it lacks a true physical reference, it is prone to overfitting as it would require extensive tuning to the specific dataset used in this study.

We therefore suggest aggregating the observation methods to hourly resolution using a simpler approach where we classify hours with any rainfall as rainy, hours with no rainfall but with snow as snowy, and hours without any precipitation as no precipitation. We have experimented with this approach, and find that for the overall evaluation of the CR and RT method, the results and conclusions are approximately the same as in the original manuscript.

As a result of the simplified aggregation method, and to other comments below, we propose to following major changes to the figures:

- **Figure 1:** We suggest showing the number of wet hours as observed by the observation methods.
- **Figure 2:** We suggest including, in addition to MCC, accuracy, precision, recall and F1 score.
- **Figure 3:** This figure shows the MCC score as a function of temperature for the RT and CR methods. As one of the reviewers highlighted, this figure does not add much more than what is already presented in Figure 2. We suggest replacing this figure with a confusion matrix, so that readers can better understand how the CR and RT classifications differ (also suggested by the same reviewer). Additionally, we propose adding a table with corresponding accuracy, precision, recall, F1, and MCC scores.

- **Figure 4:** This figure shows how the CML overestimates precipitation amounts around zero degrees Celsius compared to the radar reference. We suggest removing the left column, as we mainly discuss the right column. We also suggest removing the panels showing mixed precipitation, as the simplified aggregation method, suggested above, does not include mixed precipitation.

- **Figure 5:** This figure shows a map of the CMLs and the estimated precipitation type, including mixed precipitation. One of the reviewers suggested including more information about the topography in the map. Since we suggest not including mixed precipitation directly in our analysis, we propose changing this figure to show the interpolated number of rainy and snowy hours estimated by the CR method, RT method, and disdrometer ground truth over a two-day period. Further, by zooming out and distorting the CML coordinates, we can plot the CML locations as well as the Norwegian coastline without revealing the exact CML positions, showing larger trends and allowing for a better understanding of the climatic differences. We have experimented with this and find that the patterns and differences between the RT and CR methods become more pronounced.

- **Figure 6:** This figure shows a time series of the CR and RT method. Both of the reviewers struggled to see the benefit of adding road surface conditions and mixed precipitation, but they found the figure very useful for understanding how the CR and RT method actually works. We therefore suggest simplifying this figure by removing the road surface conditions, and move the figure to the beginning of the results section so that the readers are exposed to this figure first. Further, to make the figure more relevant we suggest to instead plot the timeline of one or more of the CMLs shown in the map (previously Figure 5). We have experimented with this, and find this approach better connects the figures, making it more engaging for the reader.

Please find our reply in blue to the individual issues raised by the reviewer.

**Review of the following manuscript**

Title: Combining commercial microwave links and weather radar for classification of dry snow and rainfall

Author(s): Erlend Øydvin et al.
MS No.: egusphere-2024-2625
MS type: Research article

This manuscript proposes simple methods to classify the precipitation type (i.e. rain/snow/dry) based on opportunistic data collected by wireless microwave links in combination with radar data (what the authors called "CR method"), relying on the different sensitivity of microwave links and weather radars to snow.

Gathering valuable information about snow from CML data has not been really addressed so far. Hence, the topic of this manuscript is interesting. Using joint radar and CML data for meteorological purposes is interesting as well. So, I think there is enough novelty and interest in this contribution. However, I see a number of points that should be addressed by

the authors before publication. Therefore, I recommend to accept the manuscript after a major revision.

**Major points**

1. Datasets. The authors put together data from 2179 CMLs across Norway, later reduced to 550 and 435 for the winter and summer datasets (Sec. 2.3). However, they didn't provide much information about the characteristics of these links (they just mentioned lower and upper bounds on lines 135-136 of their manuscript). In particular, it would be good to provide the frequency vs length distribution and the quantization error, which determine the minimum detectable rainfall intensity as well the accuracy of CML measurements. I know that CML metadata have some issues due as the owners are private companies as Ericsson, but I saw these data published in several papers on this subject. Hence, we do not know which is the sensitivity of these links to light precipitation rates, which are typical of the winter period where also snow is present in cold regions. This would be also important to understand to what extent the scatter in the data highlighted for instance in Figs. 2-and 4 is due to some/several/many CMLs performing worse than others as their sensitivity is lower, or whether it is due to the classification method itself.

We agree that the characteristics of the CMLs should be shown, and suggest to include a plot in the methods section showing this information. The quantization of the CML signal level is the same for all CMLs (0.3 dBm), except for one. We suggest removing this CML so that all CMLs have comparable quantization.

2. Methods. In particular, the way labels (i.e. precipitation type) were assigned. If I got it correctly, CML data are sampled every 1-min, radar data are available every 5-min, disdrometer data are provided every 10-min and meteorological data come every 1 hour. Due to the different sampling rate of the sensors involved, it is necessary to define a suitable time window within which labels are assigned.

   a. I would like that the authors clearly state how they put together the 1-min CML slots over the integration window (arguably 1-hour) to decide whether an hour is rain or anything else. The same should be done for the other sensors. I think these concepts are written in Sec. 3.4, but it's not the right place in my view. This is about methods. Moreover, the author should provide some justification for all those threshold values they used. Another important detail on methods is somewhat hidden as it is delayed to lines 290-292.

We agree that the way the aggregation of the different observations is described is hard to follow. Moreover, as the results can be somewhat tuned based on how the aggregation is done we suggest simplifying the aggregation method so that we classify hours with any rainfall as rainy, hours with no rainfall but with snow as snowy, and hours without any precipitation as no precipitation. We suggest to describe this aggregation in the methods section as follows:

- "Since the weather radar, temperature model, CML, and disdrometer operate on different locations and at different time resolutions, their estimates might not be synchronized, potentially leading to erroneous comparisons. Aggregating the disdrometer, CR, and RT estimates to hourly intervals can help smooth out these differences. However, some hours may contain both snow and rain, complicating the aggregation process. One solution is to introduce a mixed class, classifying hours with both snow and rain as mixed precipitation. However, it remains unclear whether

true mixed precipitation, like wet snow, would consistently cause the disdrometer, CR, and RT methods to alternate between detecting rain and snow, which could make the mixed class less physically meaningful and lead to inconsistent representations across different estimation methods. For instance, the RT method uses temperature data with hourly resolution, resulting in each hour being classified as either snowy or rainy. Mixed precipitation could be introduced by setting precipitation within a fixed temperature interval as mixed, yet the actual precipitation within this temperature interval could still be purely rain or purely snow, leading to inaccurate classifications. Additionally, since the radar might estimate precipitation slightly before the CML, the CR method is prone to estimate snow before rainfall events, leading to an overestimation of mixed precipitation. This could be addressed by aggregating the CML wet period so that the radar precipitation estimates fall inside the wet period, but this would require further tuning to avoid estimating too many rainfall events at the expense of fewer true mixed events. Another way to aggregate the estimates to hourly resolution is to drop the mixed class and classify each hour based on the most frequently estimated precipitation type. However, this approach has its own issues. True mixed precipitation, like wet snow, may not consistently show up as a mix of rain and snow in the observation methods, leading to inaccurate classifications. Additionally, this method would require fine-tuning of the disdrometers, CR, and RT methods, which adds complexity and uncertainty. Thus, to simplify the aggregation approach and ensure more consistent and accurate classifications, we have chosen to classify hours with any rainfall as rainy, hours with no rainfall but with snow as snowy, and hours without any precipitation as no precipitation for the disdrometers, CR method, and RT method. Note that switching the role of rain and snow would lower the performance of the CR method, as it often predicts snow due to the CML and radar not being synchronized."

Lines 290-292 mention that the RT method is evaluated at the CML midpoint, we agree that this information should be clearly stated in the methods section.

     b. Related to previous point a): it is not easy (at least to me) to understand how the authors assigned the precipitation type labels to get the results in Fig. 2 just based on what the authors wrote in previous Sec. 2.

We hope that the suggested simplified aggregation can better help the reader understand what is going on.

     c. Not clear which is the sampling rate of disdrometer data. On line 128 it is written that "The disdrometers […] provide an estimate of the precipitation type every 10 minutes". However in Fig.1 it seems that data are every 1-min. Please clarify.

The way this was done was to just interpolate the CML estimate, so that if the disdrometer estimated 10 minutes of snow, then snow would be assigned to every one of these minutes. However, we suggest simplifying the aggregation process, see comment above so that this will be less of an issue.

3. Results: I think the results in Sec. 3 are not presented in the best way. Reading the abstract, the purpose of this paper is clear: "This study introduces a new approach to improve rainfall and dry snow classification by combining weather radar precipitation

detection with CML signal attenuation […]. Both methods were evaluated using ground measurements from disdrometers.". Hence, I expect a basic performance assessment of the RT and CR classification methods proposed by the authors against the ground truth. However, looking at Figs. 1-6, well it's not very clear to me.

    a.  I agree MCC is a comprehensive performance indicator but it's not easy to understand how MCC values in Fig. 2 turn into good or bad labelling of data. I think a simple contingency table with indexes as Specificity and Recall for either method would help.

Good point. In addition to MCC we suggest adding accuracy, precision, recall, F1 scores as well as a confusion matrix.

    b.  Fig. 3 puts together hundreds of CMLs (which I guess have different performance as rainfall sensors). The trends in the figure can be identified looking at Fig. 2. I don't think it brings a lot of extra information. Indeed there are just a couple of short statements in the text that comment this figure.

We agree to remove this figure and replace it with a confusion matrix.

    c.  Fig. 4: I didn't get well what is shown in rows 2-4 of this figure from the explanation in the text. On lines 205-207 it is written that "In the second, third and fourth row, we have plotted the fraction of hours within the bins where the disdrometer recorded at least 10 minutes of rain, snow and both snow and rain (mix) respectively." The term "bins" is maybe inappropriate. I would use it for an histogram. Let me see if I got it. First, the counts in the top row are now hours rather than minutes. Right? Then in rows 2-4, these counts have different colours according to the fraction of time hours were flagged as rain/snow/mix by the disdrometer. So, if an hexagon in the first row has a color corresponding to 10 counts, that is 10 hours, the same point in the second row is colored according to the fraction of this 10 hours flagged as rain by the disdrometer? I don't know if I got it. Whatever is the case, please explain it as I tried to do, as it is hard to understand it from the short explanation in the text. Moreover, the colorbar is not the best. I would have used a blue scale for fractions < 0.5 and a red scale for values > 0. Having stated this, the results in Fig. 4, at least what one can notice first sight, is expected in my view: a lot of rainy time above 0°C and a lot of snowy time below 0°C and finally an uncertainty region around 0°C. Moreover, we cannot say whether a 2.5 mm/h difference in CML vs radar accumulation is large or not if we do not know the exact magnitude of rain accumulation. Looking at some finer features: it is a bit strange that radar overestimates so much in several cases at large positive Td values. Is it maybe that those were high precipitation summer events? Hence, the fractional difference Radar-CML is much less.

We acknowledge that it is hard to interpret this figure. However, by changing the aggregation method to the simpler one suggested above, we think it would be easier to understand it. We therefore suggest to simplify the figure by:
-   removing the mixed class, as we disregard the mixed class in the simplified aggregation.
-   remove column 1, as wet/dry classification by the radar is not important for understanding the RT and CR method.

    d.  Fig. 5: I see really little information here. What's the purpose of showing this figure? On the other hand, I think Fig. 6 is very useful as it shows the whole story as seen by the different sensors. I would have moved this one forward as Fig. 2 because it is very easy to understand and helps the reader in interpreting better the scatterplots. Good job here!

We suggest changing the map by zooming to a larger area and counting the number of rainy and snowy hours over a 2 day period. Focusing on a larger area will let us plot the CML distorted locations while also showing large scale trends in the distribution of snow and rainfall.

4.  Discussion. More than issues, here I am just pointing some other possible explanations of the results.
    a.  Sec. 4.1. The authors argue that disdrometer could fail as ground truth in some cases. In addition, should we 100% trust estimates of Td based on ERA5? These are not ground data measured by weather stations. Maybe it would be good to check ERA5 RH and Ta outcomes against some weather stations that are for sure available (for instance as you did in the third row of Fig. 6). Indeed, looking at Eqn. (1), Td can range between -2 and 0°C with Ta=0°C and RH ranging from 90 and 100%. It means that a 10% error on RH measurement/estimate turns into a 2°C error in Td estimate.

We agree that using data from ground based sensors would improve the RT method estimates. However, the intention of using the downscaled ERA5 estimates was that these are available everywhere, and thus provide the most realistic competitor to the CR estimates. Thus we do not think a comparison of ERA5 data to sensors on the ground is necessary. Instead we suggest adding the following to the methods section:
-   "It should be noted that model data carries inherent uncertainties due to factors like model assumptions and the downscaling process."

We also suggest adding a brief discussion on this when discussing the suggested confusion matrix (see summarizing comment) in the discussion section.

    b.  Sec. 4.2 (but also Sec. 4.1). The only way to check the effect of spatial distance between CMLs and disdrometers is to select a subset of CMLs with a disdrometer in the neighbourhood and calculate the MCC:

We have selected disdrometers within 8 km. This was a tradeoff between having enough data and not having disdrometers too far away from the CML. Since we evaluate the RT and CR estimates at similar distances from the disdrometer, the methods should have the same bias. We have experimented with plotting the MCC for different CML-disdrometer distances, and we find that the MCC only decay slightly with distance. Moreover, the CR method really only provides an improvement at temperatures around 0 degrees, making temperature a more interesting variable. We suggest not to show a figure of the metrics for different distances, as it does not directly contribute to better understanding the difference between the CR and RT method.

    c.  About wet snow. Looking at Fig. 4 wet snow seems to correlate with Td (as expected). I expect wet snow to occur at small negative values of Td, while dry snow to occur at colder temperatures. Below -5°C it's mostly dry snow

according to Fig. 4. Can the author refine a little bit their decision algorithm including a Td threshold to discriminate between wet and dry snow? Could it be that some unexpected results are due to the way data with different sampling rates were combined together in the hourly time windows and the threshold values used to flag rain/snow /mix/dry intervals)

That would indeed be interesting. Unfortunately the disdrometers do not provide estimates about dry and wet snow, making such an experiment difficult to do. In the above comments we have suggested simplifying the classification method so that the assumptions made are more explicit. This would still allow for a discussion about the effect of wet snow, but more related to the performance of the CR and RT method.

**Minor comments:**
Line 44: "Human observations can be subjective and aren't suitable for continuous high frequency monitoring" The term "high frequency" is a bit ambiguous in this context. I guess you mean high-rate monitoring.
We suggest rephrasing the line to read
  - "Human observations can be subjective and aren't suitable for continuous high-rate monitoring."

Lines 174-175: I cannot understand the statement. "Our dataset consists of CML-disdrometer pairs from the summer dataset and the winter dataset. Every minute each pair provides several different observations such as disdrometer observed precipitation type, dew point temperature and CML signal loss.". Isn't the dew point temperature taken from the ERA5 dataset? "Our dataset" in my eyes are the data "we produced ourselves", but it's not the case. Why radar maps are not part of "our dataset"?
We suggest simplifying the aggregation process, see major points 2 and 3. The radar maps are converted to rainfall rates along each CML.

Line 174: you state that disdrometer data are taken every 1-min while previously (line 128) you stated that precipitation type classification from disdrometers is available every 10-min. Please clarify.
We suggest to simplify the aggregation process so that it is easier for the reader to comprehend, see major points 2 and 3.

Lines 179-182, comment on second row of the figure. I would say "snow mostly below 0°C" while "rainfall is mainly above 0°C".
Yes. We suggest changing this Figure to include the RT and CR estimates instead.

Line 186: "using monthly time series". To me "monthly time series" means that you have used a time series long one month and extract information from the time series as a whole. Please explain-
We suggest deleting this as it does not convey any important information.

Line 187: pair instead of pairs
We agree.

Figure 2: the gray cluster is hardly visible in the two bottom panels. It would be better to use a darker tone of gray.

We suggest changing the color map to one of matplotlibs sequential colormaps

Lines 186-192 (Comments of Figure 2): I see MCC_rain of CR is usually better when it's above 0.4, maybe worth to state it-.
Yes, we further suggest adding a confusion matrix with several metrics so that these findings are more explicit.

Lines 193-196(Comments of Figure 2): the first thing I noticed looking at this figure is that CML really enhances MCC_rain wrt radar (as expected).
That is true. We suggest adding this to the discussion:
- "In terms of rainfall classification, the CR method performs just as well or outperforms the RT method for all temperatures above -2 degrees (Table 3). This could be due to the fact that the CMLs are located on the ground, which situates them closer to the disdrometers compared to the radar beam, or due to the radar beam being blocked by mountains."

Figure 5, caption: I guess it is "mix(red)" instead of "dry(red)".
We suggest simplifying the aggregation process so that mixed precipitation is not directly a part of our results. See comment above.

Line 264: 10°C instead
Ok

Line 453: DOI is not correct
Ok

---

## Author Comment (AC2)

The authors would first of all like to thank the reviewers for their thoughtful and constructive review. We note that the goal of our manuscript was to evaluate the performance of a novel method for classifying precipitation types, that works by combining estimates from weather radar and commercial microwave links (CR method), and compare it to ground truth disdrometer estimates located along roads in Norway. This method was compared to estimates from weather radar and a downscaled ERA5 temperature model (RT method) against the same disdrometers. Since the different data sources have different temporal resolutions and spatial representations, they must be aggregated in order to be comparable. Our main take away from the review was that this aggregation should be performed in a clearer and more understandable manner, and that parameters used in the CR method (length of wet period) and RT method (temperature threshold) should be better justified.

In the original manuscript the observation methods (CR method, RT method and disdrometer ground truth) were compared using 1 minute resolution and the observation methods estimates were resampled so that they could be compared at that scale. While we think that this is a valid approach, we acknowledge that it would be easier for the reader to understand the aggregation method if the observation methods were compared at 1 hour resolution, as suggested by one of the reviewers. One aspect that arises from aggregating the disdrometer data to hourly resolution is how to classify hours where there have been observed a mixture of rain and snow. An intuitive approach would be to classify these hours as mixed precipitation. However, none of the observation methods used in this experiment directly estimates true mixed precipitation, such as wet snow, and it is therefore not known how these true mixed events show up across the different observation methods. Introducing a mixed class thus requires tuning of each of the observation methods. We have experimented with this mixed class, but found that since it lacks a true physical reference, it is prone to overfitting as it would require extensive tuning to the specific dataset used in this study.

We therefore suggest aggregating the observation methods to hourly resolution using a simpler approach where we classify hours with any rainfall as rainy, hours with no rainfall but with snow as snowy, and hours without any precipitation as no precipitation. We have experimented with this approach, and find that for the overall evaluation of the CR and RT method, the results and conclusions are approximately the same as in the original manuscript.

As a result of the simplified aggregation method, and to other comments below, we propose to following major changes to the figures:

- **Figure 1:** We suggest showing the number of wet hours as observed by the observation methods.
- **Figure 2:** We suggest including, in addition to MCC, accuracy, precision, recall and F1 score.
- **Figure 3:** This figure shows the MCC score as a function of temperature for the RT and CR methods. As one of the reviewers highlighted, this figure does not add much more than what is already presented in Figure 2. We suggest replacing this figure with a confusion matrix, so that readers can better understand how the CR and RT classifications differ (also suggested by the same reviewer). Additionally, we propose adding a table with corresponding accuracy, precision, recall, F1, and MCC scores.

We have experimented with this and find that the general results and conclusions remain the same.

- **Figure 4:** This figure shows how the CML overestimates precipitation amounts around zero degrees Celsius compared to the radar reference. We suggest removing the left column, as we mainly discuss the right column. We also suggest removing the panels showing mixed precipitation, as the simplified aggregation method, suggested above, does not include mixed precipitation.

- **Figure 5:** This figure shows a map of the CMLs and the estimated precipitation type, including mixed precipitation. One of the reviewers suggested including more information about the topography in the map. Since we suggest not including mixed precipitation directly in our analysis, we propose changing this figure to show the interpolated number of rainy and snowy hours estimated by the CR method, RT method, and disdrometer ground truth over a two-day period. Further, by zooming out and distorting the CML coordinates, we can plot the CML locations as well as the Norwegian coastline without revealing the exact CML positions, showing larger trends and allowing for a better understanding of the climatic differences. We have experimented with this and find that the patterns and differences between the RT and CR methods become more pronounced.

- **Figure 6:** This figure shows a time series of the CR and RT method. Both of the reviewers struggled to see the benefit of adding road surface conditions and mixed precipitation, but they found the figure very useful for understanding how the CR and RT method actually works. We therefore suggest simplifying this figure by removing the road surface conditions, and move the figure to the beginning of the results section so that the readers are exposed to this figure first. Further, to make the figure more relevant we suggest to instead plot the timeline of one or more of the CMLs shown in the map (previously Figure 5). We have experimented with this, and find this approach better connects the figures, making it more engaging for the reader.

Please find our reply in blue to the individual issues raised by the reviewer.

Review of: "Combining commercial microwave links and weather radar for classification of dry snow and rainfall" by Erlend Øydvin et al. (2024)

This paper presents an interesting novel method to classify dry snow and rainfall by merging two precipitation measurement techniques. Previous studies has shown the difficulty to estimate precipitation phase, yet is a relatively important process in the hydrological cycle. The authors perform an elaborate analysis of their novel method and compare this with two more established methods. In general, this is a well-written manuscript. However, some parts would benefit from a revision, in order to improve the overall quality of the manuscript. Below I have stated more general and specific comments, which I hope the authors consider to be constructive.

**General:**
- Results:
  Parts of the result section are hard to understand due to the combination of complex figures in combination with it being unclear to which subplot is being referred. It would already be helpful if all subplots are labelled with a,b,c… (which might be a requirement by AMT, check their website) in combination with adding in-text

references to the specific subplots. Moreover, for the reader, especially those who are less familiar with CMLs, it would be perhaps worth considering to add a basic time-series (like Fig. 6) to the start of the results section, in order to show the reader how the methods workout in reality and help get an idea about the differences between the CR and RT method. In fact, such a figure already illustrates a lot and prepares the reader for the following more in-depth analysis. Additionally, for some figures it is not clear how you used the data to create the figure (e.g., Fig. 2, see specific comments).

We agree that the results part is too complex as it is now. We suggest simplifying the aggregation method so that the same method is used for all figures and removing some of the figures (see summarizing comment on the top). We also suggest labeling all subplots.

- Road surface conditions:
  I am struggling to see the added value of Sect. 3.5 in which the road friction is related to the classification methods (except the discussion of Fig. 6 which helps understanding the methods). I understand that the new method could also be used to estimate road conditions and I do agree that it is a nice illustration of the potential of these methods, but I think discussing road friction would require a more elaborate analysis. As you write in the discussion, road conditions are not simply a result of dry snow or rainfall, but also previous conditions (e.g., was there already a snow pack, has there been any snow/ice removal, road temperature preceding rain event) while also other precipitation phases should be considered (e.g., what about freezing rain?). Also until Sect 2.6, there hasn't been any reference of the road conditions, so it came as a bit of a surprise. Based on the introduction,I would have expected that your new method would have been applied to something like a hydrological model.

We agree that including the effect of wet snow on road surface conditions is a bit speculative without providing a larger analysis, which is tricky due to the already mentioned complexities of freezing roads. We therefore suggest removing Fig. 6 and adding simpler time series plots, that do not include road friction and road conditions, as suggested by the reviewer above.

- Discussion:
  I like that the discussion is relatively concise, but it does not include many references to previous studies on similar topics. I encourage you to include some references to previous studies, so that the reader would be better able to place your results into a wider perspective. For example, how do your findings compare with other studies that use similar methods (mostly for the RT method)? See also the specific comments.

While it would be very good to add this information, we think there are some challenges with comparing the RT estimates to other studies. We suggest to add the following text to the discussion to clarify this:
- "While other studies such as Gjertsen and Ødegaard (2005), Casellas et al. (2021), and Saltikoff et al. (2015) have evaluated the performance of temperature-based precipitation phase classification methods, these studies typically vary in methodology, terrain complexity, radar technology, and the instruments used to estimate the ground truth. Furthermore, although the disdrometers used in this study provide a large dataset, their inherent uncertainties, particularly the large number of rainy events recorded below zero degrees and the lack of a mixed class, introduce

errors that are specific to this study. These factors make it challenging to directly compare the RT estimates in this study to those from similar studies."

**Specific comments:**

L9: perhaps include the location of your study.
We suggest rephrasing the line to read:
- "Both methods were evaluated using ground measurements from disdrometers within 8 km of a CML, analyzing data from Norway using 550 CMLs in December 2021 and 435 CMLs in June 2022."

L12: There is no mention of the application to road conditions.
We suggest removing the application to road conditions as it is not important in answering our main research question related to the RT and CR method. See other comments.

L54-58: You introduce the dewpoint temperature here, but would it be an idea to include the explanation as to why both are important (L19-33) here? This will help the reader relate the dewpoint temperature to the underlying processes. Or perhaps even referring back to the previously mentioned importance of profiles would already help the reader.
We suggest changing the lines 54-56 to read:
- "Although some studies do not observe any benefit of including humidity (Leroux et al., 2023), humidity is thought to be an important parameter since the atmospheric moisture level affects the melting and evaporating precipitation, influencing whether precipitation reaches the ground as solid or liquid (Kuhn, 1987). One common measure combining humidity and temperature is the dew point temperature, which is the temperature at which air becomes saturated with water vapor at the current water content (Lawrence, 2005). The dew point temperature works by indicating the temperature at which condensation begins, reflecting the moisture content in the atmosphere and thus affecting the rate of evaporation and melting of precipitation, which influences whether it reaches the ground as solid or liquid. This measure can provide important insights into the atmospheric conditions and aids in classifying precipitation types (Feiccabrino, 2020; Harder and Pomeroy, 2013, 2014)."

L73-85: I suggest to include some more studies where they show the influence of various precipitation types on the CML signal intensity in more detail. See for example: Hansryd, J., Li, Y., Chen, J., & Ligander, P. (2010). Long term path atenuation measurement of the 71–76 GHz band in a 70/80 GHz microwave link. Proceedings of the Fourth European Conference on Antennas and Propagation van Leth, T. C., Overeem, A., Leijnse, H., & Uijlenhoet, R. (2018). A measurement campaign to assess sources of error in microwave link rainfall estimation. Atmospheric Measurement Techniques, 11(8), 4645-4669. htps://doi.org/10.5194/amt-11-4645-2018
Good suggestion, we propose adding the following to L80:
- " Only a few studies have focused on how CMLs are affected by colder climates. Hansryd et al. (2010) reported that dry snow caused minimal signal attenuation, while wet snow caused higher signal attenuation. van Leth et al. (2018) observed that during an event with a mixture of rain and snow, the CMLs experiences a strong signal attenuation which persisted for about 10 minutes after the snowfall even"

L79-80: Based on this line, I would have expected the paper to focus on applying/elaborating on the methods of Cherkassky and Ostrometzky. I suggest to rephrase this.
We suggest removing this line and merge the following paragraph with the paragraph above.

L83: reports  reported, in order to use the same tense as the previous references (very minor comment)
We agree.

L90: I would suggest to already mention here that you cannot share the location of the CMLs due to data security reasons.
We suggest adding the following to L90:
  - "According to the agreement with Ericsson, the exact location of the CMLs are secret and cannot be shared."

L103-104 (and L118-120): I struggle to understand why you do this. As I understand you extend the wet periods in order to make sure that the rain event has also completely passed the disdrometer, is that correct? Yet in your analysis, there is no comparison of rain events but you compare individual time steps and average/summed periods, so I don't see how this period extension exactly works. Or am I missing something here?
This is a valid remark, in short the idea is that by extending the wet periods, there is a higher chance that the predicted wet period will coincide with the true wet period observed by the disdrometer. However, following the review we suggest simplifying this aggregation by instead setting hours with any rainfall as rainy, hours with no rainfall but with snow as snowy, and hours without any precipitation as no precipitation for the disdrometers, CR method, and RT method. See major comment 2a by reviewer 2 and summarizing comment on the top.

L107: Could you explain briefly on what concept the method of Leijnse et al. (2008) is based? Not every reader will know how their method works.
Since we suggest not doing any CML precipitation estimates in the revised article, we suggest removing this sentence.

L110-114: Could you provide a reference to this radar product for readers who would be interested to know more about this product?
There are unfortunately not any official references to the radar product.

L113-115: Has the seacluter and other large peaks been removed by you or was that already done in the data you downloaded?
It was already removed.

L121-136: How well do these disdrometers work? Are there any known biases or uncertainties? Especially because in Sect. 4.2 you refer to their potential uncertainty. This could also be discussed in the discussion.
There is, as far as we are aware, no official experiment indicating how well the disdrometers employed by the road authorities work. As far as we know, this is the first official work investigating the disdrometer performance.

L132: Why do you use 8km? Is there a specific reason for this? Is this based on previous studies, or based on your own data? Or is this a common value in CML studies?

This is just picking a number that gives a reasonably large number of CML-disdrometer pairs, without going too far from the CMLs.

L138: Are there any common biases or uncertainties in this data? If so, please mention to help the reader.
As this is downscaled ERA5 model data there are indeed uncertainties in this dataset. We suggest adding the following sentence to L138:
- "It should be noted that model data carries inherent uncertainties due to factors like model assumptions and the downscaling process."

L147: Marks et al. use 0 C, but that is for the USA right? Why would that be applicable to Norway? As you write in your introduction, these threshold values can have a relatively large range, and thus can have a relatively large influence on your results.
You are correct that the threshold would vary based on different locations. It can also change during the precipitation events. Thus, while the RT algorithm could be tuned to match the observed precipitation type, we think that doing this would provide an unfair advantage of the RT method over the CR method. Moreover, the purpose of the RT method is mainly to provide a reliable reference method so that the errors of the CR method can be better understood. We have therefore used a threshold from the literature.

L155: Perhaps show the equation for MCC. I think not all readers will be familiar with it.
We agree, and following the review, we suggest adding the full confusion matrix as well as several other metrics. See summarizing comment on the top.

Fig 2.: The caption is hard to understand here, because you refer to RT and the disdrometers both as references. Also, it is not clear what you mean with "the MCC's are computed for each CML-disdrometer pairs using 1 month of data". What do you mean with 1 month of data? Just the December data? It almost seems like there is an additional step between the data described in Sect. 2 and this figure. (also minor comment: I think it should be pair instead of pairs)
We suggest rephrasing the Figure caption to read
- "Scatter density plots (a, b, c, d, e) comparing the accuracy, precision, recall, F1 and MCC score for the CR and RT method for each CML-disdrometer pair. Average temperature of each cell (f, g, h, i, j)"

Fig. 3: Is this the average of the MCC's for all CML-disdrometer pairs as a function of Td?
Yes, however, we suggest replacing this figure by a confusion matrix to address the review, see summarizing comment.

L198-210: Here the addition of in-text references to subplots a,b,etc. would help to guide the reader.
We suggest removing this Figure as, following reviewer 2, it could be replaced by a confusion matrix.

L211-221: I got confused here because of the text in the parentheses. Why are both the RT and the CR method referred to as CML rain/snow/dry? Wouldn't it be an idea to just refer to them as rain/snow/dry? Or is there a specific reason you included CML?

The difference between the left and right column is that in the left column, the CML rainfall rate was derived from wet periods as classified by weather radar, while in the right we used a CML based wet-dry detection method. We suggest instead using only CML wet detection (keep the right column) as using both radar and CML to identify wet periods is not necessary.

L214: to be fully correct the radar observes precipitation instead of rain (very minor comment)
We agree that this could be stated more clearly.

Fig 5. I understand that you cannot share the exact location of the CMLs because of data security reasons, but is there the possibility to describe the landscape a bit? For example, are there any large elevation differences (i.e., mountiains/fiords) in this area that could for example create orographic precipitation or cause beam blockage of the radar? Additionally, I suggest to add titles to the columns, so that it is immediately clear which method is shown in which column.
Adding the location of the CMLs and indicating the climatological differences would indeed make the article more engaging. We therefore suggest changing the map by focusing on a larger area and distort the CML coordinates slightly so that the climatological differences are visible, but the CML coordinates are secret.

Discussion: Are there any previous studies in which the RT method (or something similar) has been compared with disdrometers (or other precipitation phase observations). If so, I would recommend to include a short discussion on this. How do your results compare to those studies? Are they similar or is it the RT method more difficult in Norway because of for example the elevation differences? Such a discussion would allow to put the performance of the CR method into a wider context.
This is an important remark. We have addressed this remark above in the major comments.

L264: the circle of degree Celsius should be in superscript (very minor comment)
We agree.

L266: How often does a disdrometer misclassify precipitation phase? I would advise you to include a brief discussion based on previous literature regarding uncertainty when using disdrometers to estimate precipitation phase.
We suggest adding the following to the discussion
- "Disdrometers do not provide 100% accurate records of precipitation type. There is notable variability in agreement between identical disdrometer models, which agree about 90% of the time, and this variability tends to increase when different models are compared (Pickering et al., 2021; Friedrich et al., 2013; Bloemink, 2005)"

Fig 6: In the bottom subplot, I suggest to use a different shading color for the diff<-5 mm, because it overlaps with the shadings above. Or perhaps you can even leave this out, as you do not discuss this in the text if I'm correct.
We suggest removing this plot and replace it with simpler plots showing CML time series.

L270-285: Based on this section, it seems that only wet snow could cause misclassifications. I would suggest to make clear that any form of wet precipitation causes the CML signal to

drop. Additionally could it also be that spatial temperature and humidity differences can cause that at the disdrometer location dry snow is falling while somewhere else the precipitation has started melting? I can imagine that this happens when temperatures are in the transition region between rain and snow.

To clarify this we suggest adding the following to the discussion
- "We note that although this study focused on rain and snow, any type of wet precipitation can cause the CML signal level to drop, leading to potential misclassifications. Another source of uncertainty is the spatial distance between the disdrometer and the CML midpoint, where temperature variations due to elevation and spatial differences can affect precipitation classification."

L288-289: Are there any previous studies that show differences between radar and CML/disdrometer measurements? If so, I suggest to discuss these here.

See comment to the 3rd general comment.

---

## Author Response (AR1)

General remarks

The authors would first of all like to thank the reviewers for their thoughtful and constructive review. Responses to their specific comments are given further down. As the manuscript has undergone quite a bit of change, we provide an initial overview of those changes here.

We note that the goal of our manuscript was to evaluate the performance of a novel method for classifying precipitation types, that works by combining estimates from weather radar and commercial microwave links (CR method), and compare it to ground truth disdrometer estimates located along roads in Norway. This method was compared to estimates from weather radar and a downscaled ERA5 temperature model (RT method) against the same disdrometers. Since the different data sources have different temporal resolutions and spatial representations, they must be aggregated in order to be comparable. Our main take away from the review was that this aggregation should be performed in a more transparent manner, and that parameters used in the CR method and RT method should be better justified.

In the original manuscript the observation methods (CR method, RT method and disdrometer ground truth) were compared using 1 minute resolution and the observation methods estimates were resampled so that they could be compared at that scale. While we think that this is a valid approach, we acknowledge that it would be easier for the reader to understand the aggregation method if the observation methods were compared at 1 hour resolution, as suggested by one of the reviewers. One aspect that arises from aggregating the disdrometer data to hourly resolution is how to classify hours where there have been observed a mixture of rain and snow. An intuitive approach would be to classify these hours as mixed precipitation. However, none of the observation methods used in this experiment directly estimates true mixed precipitation, such as wet snow, and it is therefore not known how these true mixed events show up across the different observation methods. Introducing a mixed class thus requires tuning of each of the observation methods. We have experimented with this mixed class, but found that since it lacks a true physical reference, it is prone to overfitting as it would require extensive tuning to the specific dataset used in this study.

In the revised manuscript, we have therefore aggregated the observation methods to hourly resolution using a simpler approach where we classify hours with any rainfall as rainy, hours with no rainfall but with snow as snowy, and hours without any precipitation as no precipitation. We have experimented with this approach, and find that for the overall evaluation of the CR and RT method, the results and conclusions are about the same as in the original manuscript.

Due to the simplified aggregation method and other reviewer comments, the figures have changed accordingly:

-   **Figure 1:** This figure presented the number of observations by the disdrometers, radar and CML. In the revised manuscript it shows the number of wet hours as observed by the observation methods (CR, RT and disdrometers)). New name: Fig. 6.

- **Figure 2:** This figure presented the CR and RT method MCC score of different CML-disdrometer pairs. The revised figure includes, in addition to MCC, accuracy, precision, recall and F1 score. New name: Fig. 8.
- **Figure 3:** This figure presented the MCC score as a function of temperature for the RT and CR methods. As one of the reviewers highlighted, it does not add much more than what is already presented in Figure 2. In the revised manuscript this figure is replaced by a confusion matrix, so that readers can better understand how the CR and RT classifications differ (also suggested by the same reviewer). Additionally, we have added a table with corresponding accuracy, precision, recall, F1, and MCC scores. New name: Fig. 9.
- **Figure 4:** This figure presented how the CML overestimates precipitation amounts around zero degrees compared to the radar reference. In the revised manuscript, we have removed the left column as it caused some confusion and does not add much information. We also removed the panels showing mixed precipitation, as the simplified aggregation method does not include mixed precipitation. New name: Fig. 7.
- **Figure 5:** This figure presented a map of the CMLs and the estimated precipitation type, including mixed precipitation. Since the simplified aggregation method does not consider mixed precipitation, the revised figure shows the interpolated number of rainy and snowy hours estimated by the CR method, RT method, and disdrometer ground truth over a two-day period. We also plot the distorted CML locations as well as the Norwegian coast, as suggested by one of the reviewers. New name: Fig. 2.
- **Figure 6:** This figure presented a time series of the CR and RT method. Both of the reviewers struggled to see the benefit of adding road surface conditions and mixed precipitation, but they found the figure very useful for understanding how the CR and RT method actually works. In the revised manuscript we have simplified this figure and replaced it by three CML timelines, located in the revised map. New name: Fig. 3, 4 and 5.

Please find our reply in blue to the individual issues raised by the reviewers.

**Reviewer 1**

Review of: "Combining commercial microwave links and weather radar for classification of dry snow and rainfall" by Erlend Øydvin et al. (2024)

This paper presents an interesting novel method to classify dry snow and rainfall by merging two precipitation measurement techniques. Previous studies has shown the difficulty to estimate precipitation phase, yet is a relatively important process in the hydrological cycle. The authors perform an elaborate analysis of their novel method and compare this with two more established methods. In general, this is a well-written manuscript. However, some parts would benefit from a revision, in order to improve the overall quality of the manuscript. Below I have stated more general and specific comments, which I hope the authors consider to be constructive.

**General:**
- Results:

Parts of the result section are hard to understand due to the combination of complex figures in combination with it being unclear to which subplot is being referred. It would already be helpful if all subplots are labelled with a,b,c… (which might be a requirement by AMT, check their website) in combination with adding in-text references to the specific subplots. Moreover, for the reader, especially those who are less familiar with CMLs, it would be perhaps worth considering to add a basic time-series (like Fig. 6) to the start of the results section, in order to show the reader how the methods workout in reality and help get an idea about the differences between the CR and RT method. In fact, such a figure already illustrates a lot and prepares the reader for the following more in-depth analysis. Additionally, for some figures it is not clear how you used the data to create the figure (e.g., Fig. 2, see specific comments).

We agree that the results part is too complex as it is now. In the revised manuscript we have simplified the aggregation method so that the same method is used for all figures and removed some of the figures (see summarizing comment on the top). The aggregation method is explained in chapter 2.4 (L155-178). We have also labeled all subplots according to AMT requirements.

- Road surface conditions:
  I am struggling to see the added value of Sect. 3.5 in which the road friction is related to the classification methods (except the discussion of Fig. 6 which helps understanding the methods). I understand that the new method could also be used to estimate road conditions and I do agree that it is a nice illustration of the potential of these methods, but I think discussing road friction would require a more elaborate analysis. As you write in the discussion, road conditions are not simply a result of dry snow or rainfall, but also previous conditions (e.g., was there already a snow pack, has there been any snow/ice removal, road temperature preceding rain event) while also other precipitation phases should be considered (e.g., what about freezing rain?). Also until Sect 2.6, there hasn't been any reference of the road conditions, so it came as a bit of a surprise. Based on the introduction,I would have expected that your new method would have been applied to something like a hydrological model.

We agree that including the effect of wet snow on road surface conditions is a bit speculative without providing a larger analysis, which is tricky due to the already mentioned complexities of freezing roads. In the revised manuscript we have removed the road friction case study, and replaced Fig. 6 by three simpler time series of CMLs located in the revised map. We have also moved these time series and the map to the beginning of the discussion so that the reader gets a more practical introduction to the CR and RT method early on.

Regarding applying the CR method to a hydrological model we consider this out of scope of this work and have added the following to the conclusion (L390):
- " Future work should also investigate how the CR estimates impact hydrological models."

- Discussion:
  I like that the discussion is relatively concise, but it does not include many references to previous studies on similar topics. I encourage you to include some references to previous studies, so that the reader would be better able to place your results into a wider perspective. For example, how do your findings compare with other studies

that use similar methods (mostly for the RT method)? See also the specific comments.

While it would be very good to add this information, we think there are some challenges with comparing the RT estimates to other studies. To clarify this, we have added following text to the discussion (L364-374):

- "While studies such as Gjertsen and Ødegaard (2005), Casellas et al. (2021), and Saltikoff et al. (2015) have evaluated the performance of temperature-based precipitation phase classification methods, these studies typically vary in methodology, terrain complexity, radar technology, and the instruments used to estimate the ground truth. This variability introduces challenges when comparing results across different studies. Although the disdrometers used in this study provide a large dataset, they have some limitations; in particular, the large number of rainy events recorded below zero degrees and the lack of mixed class classification introduce uncertainties specific to this study. Further, any type of wet precipitation can cause the CML signal level to drop, potentially leading to precipitation being falsely classified as rainfall. Another source of uncertainty is the spatial distance between the disdrometer and the CML, where for instance temperature variations due to elevation and spatial differences can affect precipitation classification. Moreover, the temperature model used is based on model data, and could be improved by using ground based sensors. Combined, these factors introduce large uncertainties in this study, and further make it challenging to directly compare the CR and RT estimates in this study to those from similar studies."

**Specific comments:**

L9: perhaps include the location of your study.
We have rephrased L9 to read:

- "Both methods were evaluated using measurements from disdrometers located within 8 km of a CML, taken as ground truth. The analysis used data from Norway, including 550 CMLs in December 2021 and 435 CMLs in June 2022."

L12: There is no mention of the application to road conditions.
In the revised manuscript we have removed the application to road conditions as it is not important in answering our main research question related to the RT and CR method. See also reply to reviewer 2 comments.

L54-58: You introduce the dewpoint temperature here, but would it be an idea to include the explanation as to why both are important (L19-33) here? This will help the reader relate the dewpoint temperature to the underlying processes. Or perhaps even referring back to the previously mentioned importance of profiles would already help the reader.
In the revised manuscript we have changed the lines 55-59 to read:

- "Although some studies do not observe any benefit of including humidity (Leroux et al., 2023), humidity is thought to be an important parameter since the atmospheric moisture level affects the melting and evaporating precipitation, influencing whether precipitation reaches the ground as solid or liquid (Kuhn, 1987). One common measure combining humidity and temperature is the dew point temperature, which is the temperature at which the air would become saturated with water vapor at constant pressure and moisture content (Lawrence, 2005). In a dry atmosphere, the

dew point temperature is significantly lower than the air temperature. In these conditions, melting is not favored and snow can be observed at positive air temperatures. Conversely, the dew point and the air temperatures are equal if the air is saturated, and solid precipitation will more likely have melted before reaching the ground if the air temperature is above 0 degrees (Feiccabrino, 2020; Harder and Pomeroy, 2013, 2014)."

L73-85: I suggest to include some more studies where they show the influence of various precipitation types on the CML signal intensity in more detail. See for example: Hansryd, J., Li, Y., Chen, J., & Ligander, P. (2010). Long term path atenuation measurement of the 71–76 GHz band in a 70/80 GHz microwave link. Proceedings of the Fourth European Conference on Antennas and Propagation van Leth, T. C., Overeem, A., Leijnse, H., & Uijlenhoet, R. (2018). A measurement campaign to assess sources of error in microwave link rainfall estimation. Atmospheric Measurement Techniques, 11(8), 4645-4669. htps://doi.org/10.5194/amt-11-4645-2018

Good suggestion, in the revised manuscript we have added the following to L85:
- " Other studies have focused on how CMLs are affected by colder climates. Hansryd et al. (2010) reported that heavy snowfall caused minimal signal attenuation, while a mix of rain and snow, caused higher signal attenuation. van Leth et al. (2018) observed that during an event with a mixed precipitation the CMLs experience a strong signal attenuation which persisted for about 10 minutes after the precipitation event, possibly due to melting of snow off from the antenna cover."

L79-80: Based on this line, I would have expected the paper to focus on applying/elaborating on the methods of Cherkassky and Ostrometzky. I suggest to rephrase this.
We have removed this line and merged the next paragraph with the paragraph before.

L83: reports  reported, in order to use the same tense as the previous references (very minor comment)
We have changed it to "reported".

L90: I would suggest to already mention here that you cannot share the location of the CMLs due to data security reasons.
In the revised manuscript we show the (distorted) location of the CMLs in a map (Fig. 2), which gives the reader a better idea of the spatial distribution. In the data availability section at the end of the manuscript we explain the agreement with Ericsson.

L103-104 (and L118-120): I struggle to understand why you do this. As I understand you extend the wet periods in order to make sure that the rain event has also completely passed the disdrometer, is that correct? Yet in your analysis, there is no comparison of rain events but you compare individual time steps and average/summed periods, so I don't see how this period extension exactly works. Or am I missing something here?
This is a valid remark, in short the idea is that by extending the wet periods, there is a higher chance that the predicted wet period will coincide with the true wet period observed by the disdrometer. However, following the review we have simplified this aggregation by instead setting hours with any rainfall as rainy, hours with no rainfall but with snow as snowy, and hours without any precipitation as no precipitation for the disdrometers, CR method, and RT method. See also major comment 2a by reviewer 2 and summarizing comment on the top.

L107: Could you explain briefly on what concept the method of Leijnse et al. (2008) is based? Not every reader will know how their method works.
In the revised manuscript we have changed L112-L117 to read
- "For the classification of rainy periods, we used the convolutional neural network (CNN) presented by Polz et al. (2020), as this model was trained on hourly data and this study also evaluates estimates at an hourly resolution. The baseline was estimated by using the average signal attenuation 5 minutes before a rainy period. Water droplets forming on the antenna cover cause additional attenuation, which we accounted for by using the semi-empirical wet antenna attenuation model proposed by Leijnse et al. (2008), with the refined parameters suggested by Graf et al. (2020). Finally, the rainfall rate was computed using the k-R relation, with parameters defined by ITU (2005)."

L110-114: Could you provide a reference to this radar product for readers who would be interested to know more about this product?
There are unfortunately not any official references to the radar product.

L113-115: Has the seacluter and other large peaks been removed by you or was that already done in the data you downloaded?
It was already removed by the MET office. We tried with raw radar data, but there is a lot of clutter that needs special treatment. Unfortunately, the exact procedure is not documented and we think documenting this is beyond the scope of this paper.

L121-136: How well do these disdrometers work? Are there any known biases or uncertainties? Especially because in Sect. 4.2 you refer to their potential uncertainty. This could also be discussed in the discussion.
There is, as far as we are aware, no official experiment indicating how well the disdrometers employed by the road authorities work. As far as we know, this is the first publicly available official work investigating the disdrometer performance. In the revised manuscript we have added a discussion about the disdrometer performance and included some notes about the general performance of disdrometers (L280-L302).

L132: Why do you use 8km? Is there a specific reason for this? Is this based on previous studies, or based on your own data? Or is this a common value in CML studies?
This is just picking a number that gives a reasonably large number of CML-disdrometer pairs, without going too far from the CMLs. To clarify this we have added the following to L141:
- "The threshold of 8 km was a trade-off between getting a large number of CML-disdrometer pairs, while still maintaining good correlation between the pairs."

L138: Are there any common biases or uncertainties in this data? If so, please mention to help the reader.
As this is downscaled ERA5 model data there are indeed uncertainties in this dataset. In the revised manuscript (L148) we have added the following:
- "It should be noted that model data carries inherent uncertainties due to factors like model assumptions and the downscaling process."

L147: Marks et al. use 0 C, but that is for the USA right? Why would that be applicable to Norway? As you write in your introduction, these threshold values can have a relatively large range, and thus can have a relatively large influence on your results.
You are correct that the threshold would vary based on different locations. It can also change during the precipitation events. Thus, while the RT algorithm could be tuned to match the observed precipitation type, we think that doing this would provide an unfair advantage of the RT method over the CR method. Moreover, the purpose of the RT method is mainly to provide a reliable reference method so that the errors of the CR method can be better understood. We have therefore used a threshold from the literature.

L155: Perhaps show the equation for MCC. I think not all readers will be familiar with it.
We agree, and following the review, in the revised manuscript we have added the full confusion matrix as well as several other metrics (L180-219). See also summarizing comment on the top.

Fig 2.: The caption is hard to understand here, because you refer to RT and the disdrometers both as references. Also, it is not clear what you mean with "the MCC's are computed for each CML-disdrometer pairs using 1 month of data". What do you mean with 1 month of data? Just the December data? It almost seems like there is an additional step between the data described in Sect. 2 and this figure. (also minor comment: I think it should be pair instead of pairs)
In the revised manuscript we have rephrased the Figure caption to read (Fig. 8, L287-L288):
   -   "Scatter density plots **(a, b, c, d, e)** comparing the accuracy, precision, recall, F1 and MCC score for the CR and RT method for each CML-disdrometer pair. Average temperature of each cell **(f, g, h, i, j)"**

Fig. 3: Is this the average of the MCC's for all CML-disdrometer pairs as a function of Td?
Yes, however, in the revised manuscript we have replaced this figure by a confusion matrix, see summarizing comment at the top.

L198-210: Here the addition of in-text references to subplots a,b,etc. would help to guide the reader.
In the revised manuscript we have added in-text-references to the subplots.

L211-221: I got confused here because of the text in the parentheses. Why are both the RT and the CR method referred to as CML rain/snow/dry? Wouldn't it be an idea to just refer to them as rain/snow/dry? Or is there a specific reason you included CML?
The difference between the left and right column is that in the left column, the CML rainfall rate was derived from wet periods as classified by weather radar, while in the right we used a CML based wet-dry detection method. In the revised manuscript we use only CML wet detection (keep the right column) as using both radar and CML to identify wet periods is not necessary (see Fig 7 L287-L288).

L214: to be fully correct the radar observes precipitation instead of rain (very minor comment)
In the revised manuscript this text was changed.

Fig 5. I understand that you cannot share the exact location of the CMLs because of data security reasons, but is there the possibility to describe the landscape a bit? For example, are there any large elevation differences (i.e., mountains/fiords) in this area that could for example create orographic precipitation or cause beam blockage of the radar? Additionally, I suggest to add titles to the columns, so that it is immediately clear which method is shown in which column.

Adding the location of the CMLs and indicating the climatological differences would indeed make the article more engaging. In the revised manuscript we have changed the map by focusing on a larger area and distorted the CML coordinates slightly so that the climatological differences are visible, but the CML coordinates are secret (Fig. 2, L219-L220). This lets us plot the Norwegian coastline as well as describe the different climatic zones.

Discussion: Are there any previous studies in which the RT method (or something similar) has been compared with disdrometers (or other precipitation phase observations). If so, I would recommend to include a short discussion on this. How do your results compare to those studies? Are they similar or is it the RT method more difficult in Norway because of for example the elevation differences? Such a discussion would allow to put the performance of the CR method into a wider context.

This is an important remark. We have addressed this remark above in the major comments.

L264: the circle of degree Celsius should be in superscript (very minor comment)

In the revised manuscript we have fixed this.

L266: How often does a disdrometer misclassify precipitation phase? I would advise you to include a brief discussion based on previous literature regarding uncertainty when using disdrometers to estimate precipitation phase.

In the revised manuscript we have added the following to the discussion (L292):
  - "One explanation for this overestimation could be that wet snow makes the disdrometer alternate between snow and rain, creating the impression that there is more rain below zero degrees. Moreover, different precipitation type sensors are also known to disagree during mixed precipitation events (Bloemink, 2005; Pickering et al., 2021), indicating that these events are hard to classify."

Fig 6: In the bottom subplot, I suggest to use a different shading color for the diff<-5 mm, because it overlaps with the shadings above. Or perhaps you can even leave this out, as you do not discuss this in the text if I'm correct.

In the revised manuscript we have removed this plot and replaced it with simpler plots showing CML time series (Fig 3 - 5).

L270-285: Based on this section, it seems that only wet snow could cause misclassifications. I would suggest to make clear that any form of wet precipitation causes the CML signal to drop. Additionally could it also be that spatial temperature and humidity differences can cause that at the disdrometer location dry snow is falling while somewhere else the precipitation has started melting? I can imagine that this happens when temperatures are in the transition region between rain and snow.

To clarify this we have added a longer discussion of the uncertainties of this study. Specifically L371 state:

- " Further, any type of wet precipitation can cause the CML signal level to drop, potentially leading to precipitation being falsely classified as rainfall."

We also include a discussion on the spatial differences between the disdrometer and CML, for instance in this line:
- "For example, looking at the CML time series (Fig. 3), the disdrometer shows a mix of rain and snow, while the CR and RT methods indicate only snow. This discrepancy could be due to the spatial distance between disdrometers and CMLs, with temperature differences at these locations possibly causing rain to be recorded at colder temperature."

L288-289: Are there any previous studies that show differences between radar and CML/disdrometer measurements? If so, I suggest to discuss these here.
See the response to the 3rd general comment.

**Reviewer 2**

**Review of the following manuscript**

Title: Combining commercial microwave links and weather radar for classification of dry snow and rainfall

Author(s): Erlend Øydvin et al.
MS No.: egusphere-2024-2625
MS type: Research article

This manuscript proposes simple methods to classify the precipitation type (i.e. rain/snow/dry) based on opportunistic data collected by wireless microwave links in combination with radar data (what the authors called "CR method"), relying on the different sensitivity of microwave links and weather radars to snow.

Gathering valuable information about snow from CML data has not been really addressed so far. Hence, the topic of this manuscript is interesting. Using joint radar and CML data for meteorological purposes is interesting as well. So, I think there is enough novelty and interest in this contribution. However, I see a number of points that should be addressed by the authors before publication. Therefore, I recommend to accept the manuscript after a major revision.

**Major points**
1. Datasets. The authors put together data from 2179 CMLs across Norway, later reduced to 550 and 435 for the winter and summer datasets (Sec. 2.3). However, they didn't provide much information about the characteristics of these links (they just mentioned lower and upper bounds on lines 135-136 of their manuscript). In particular, it would be good to provide the frequency vs length distribution and the quantization error, which determine the minimum detectable rainfall intensity as well the accuracy of CML measurements. I know that CML metadata have some issues

due as the owners are private companies as Ericsson, but I saw these data published in several papers on this subject. Hence, we do not know which is the sensitivity of these links to light precipitation rates, which are typical of the winter period where also snow is present in cold regions. This would be also important to understand to what extent the scatter in the data highlighted for instance in Figs. 2-and 4 is due to some/several/many CMLs performing worse than others as their sensitivity is lower, or whether it is due to the classification method itself.

We agree that the characteristics of the CMLs should be shown. In the revised manuscript we have included a plot in the methods section showing this information (Fig. 1). The quantization of the CML signal level is the same for all CMLs (0.3 dBm), except for one. We removed this CML so that all CMLs have comparable quantization.

2. Methods. In particular, the way labels (i.e. precipitation type) were assigned. If I got it correctly, CML data are sampled every 1-min, radar data are available every 5-min, disdrometer data are provided every 10-min and meteorological data come every 1 hour. Due to the different sampling rate of the sensors involved, it is necessary to define a suitable time window within which labels are assigned.
   a. I would like that the authors clearly state how they put together the 1-min CML slots over the integration window (arguably 1-hour) to decide whether an hour is rain or anything else. The same should be done for the other sensors. I think these concepts are written in Sec. 3.4, but it's not the right place in my view. This is about methods. Moreover, the author should provide some justification for all those threshold values they used. Another important detail on methods is somewhat hidden as it is delayed to lines 290-292.

We agree that the way the aggregation of the different observations is described is hard to follow. Moreover, as the results can be somewhat tuned based on how the aggregation is done we have simplified the aggregation method so that we classify hours with any rainfall as rainy, hours with no rainfall but with snow as snowy, and hours without any precipitation as no precipitation. See chapter 2.4 (L155-L178) for a detailed description.

Lines 290-292 mention that the RT method is evaluated at the CML midpoint, we agree that this information should be clearly stated in the methods section and have added the following to L167:
   - "The RT method then works by estimating the average precipitation along the CML, and classifying precipitation above a dew point temperature threshold as rain and below the threshold as snow. The dew point temperature was evaluated at the pixel closest to the center of the CML and was, in line with Marks et al. (2013), set to 0 ∘C."

   b. Related to previous point a): it is not easy (at least to me) to understand how the authors assigned the precipitation type labels to get the results in Fig. 2 just based on what the authors wrote in previous Sec. 2.

In the revised manuscript we have simplified the aggregation method so that all figures use the same method. We think this makes it easier to interpret the figures.

   c. Not clear which is the sampling rate of disdrometer data. On line 128 it is written that "The disdrometers […] provide an estimate of the precipitation type every 10 minutes". However in Fig.1 it seems that data are every 1-min. Please clarify.

The way this was done was to just interpolate the CML estimate, so that if the disdrometer estimated 10 minutes of snow, then snow would be assigned to every one of these minutes. However, in the revised manuscript we have simplified the aggregation method so that all figures use the same method. We think this makes it easier to interpret the figures.

3. Results: I think the results in Sec. 3 are not presented in the best way. Reading the abstract, the purpose of this paper is clear: "This study introduces a new approach to improve rainfall and dry snow classification by combining weather radar precipitation detection with CML signal attenuation […]. Both methods were evaluated using ground measurements from disdrometers.". Hence, I expect a basic performance assessment of the RT and CR classification methods proposed by the authors against the ground truth. However, looking at Figs. 1-6, well it's not very clear to me.
   a. I agree MCC is a comprehensive performance indicator but it's not easy to understand how MCC values in Fig. 2 turn into good or bad labelling of data. I think a simple contingency table with indexes as Specificity and Recall for either method would help.

Good point. In the revised manuscript we have added accuracy, precision, recall, F1 scores as well as a confusion matrix. See Fig. 9 and Table 3 (L187-L188).

   b. Fig. 3 puts together hundreds of CMLs (which I guess have different performance as rainfall sensors). The trends in the figure can be identified looking at Fig. 2. I don't think it brings a lot of extra information. Indeed there are just a couple of short statements in the text that comment this figure.

In the revised manuscript we have removed this figure and replaced it with a confusion matrix (Fig. 9).

   c. Fig. 4: I didn't get well what is shown in rows 2-4 of this figure from the explanation in the text. On lines 205-207 it is written that "In the second, third and fourth row, we have plotted the fraction of hours within the bins where the disdrometer recorded at least 10 minutes of rain, snow and both snow and rain (mix) respectively." The term "bins" is maybe inappropriate. I would use it for an histogram. Let me see if I got it. First, the counts in the top row are now hours rather than minutes. Right? Then in rows 2-4, these counts have different colours according to the fraction of time hours were flagged as rain/snow/mix by the disdrometer. So, if an hexagon in the first row has a color corresponding to 10 counts, that is 10 hours, the same point in the second row is colored according to the fraction of this 10 hours flagged as rain by the disdrometer? I don't know if I got it. Whatever is the case, please explain it as I tried to do, as it is hard to understand it from the short explanation in the text. Moreover, the colorbar is not the best. I would have used a blue scale for fractions < 0.5 and a red scale for values > 0. Having stated this, the results in Fig. 4, at least what one can notice first sight, is expected in my view: a lot of rainy time above 0°C and a lot of snowy time below 0°C and finally an uncertainty region around 0°C. Moreover, we cannot say whether a 2.5 mm/h difference in CML vs radar accumulation is large or not if we do not know the exact magnitude of rain accumulation. Looking at some finer features: it is a bit strange that radar overestimates so much in several cases at large positive Td values. Is it maybe that those were high

precipitation summer events? Hence, the fractional difference Radar-CML is much less.

We acknowledge that it is hard to interpret this figure. However, by changing the aggregation method to the simpler one suggested above, we think it would be easier to understand it. In the revised manuscript we have (Fig. 7, L287-L288)
- removed the mixed class, as we disregard the mixed class in the simplified aggregation.
- removed column 1, as wet/dry classification by the radar is not important for understanding the RT and CR method.

      d. Fig. 5: I see really little information here. What's the purpose of showing this figure? On the other hand, I think Fig. 6 is very useful as it shows the whole story as seen by the different sensors. I would have moved this one forward as Fig. 2 because it is very easy to understand and helps the reader in interpreting better the scatterplots. Good job here!

In the revised manuscript we have changed the map by zooming to a larger area and counted the number of rainy and snowy hours over a 2 day period. Focusing on a larger area lets us plot the CML distorted locations while still showing large scale trends in the distribution of rainfall and snow. See Fig 2, L219-L220.

4. Discussion. More than issues, here I am just pointing some other possible explanations of the results.
      a. Sec. 4.1. The authors argue that disdrometer could fail as ground truth in some cases. In addition, should we 100% trust estimates of Td based on ERA5? These are not ground data measured by weather stations. Maybe it would be good to check ERA5 RH and Ta outcomes against some weather stations that are for sure available (for instance as you did in the third row of Fig. 6). Indeed, looking at Eqn. (1), Td can range between -2 and 0°C with Ta=0°C and RH ranging from 90 and 100%. It means that a 10% error on RH measurement/estimate turns into a 2°C error in Td estimate.

We agree that using data from ground based sensors would improve the RT method estimates. However, the intention of using the downscaled ERA5 estimates was that these are available everywhere, and thus provide the most realistic competitor to the CR estimates. Thus we do not think a comparison of ERA5 data to sensors on the ground is necessary. Instead we have added the following to the methods section (L149):
- "It should be noted that model data carries inherent uncertainties due to factors like model assumptions and the downscaling process."

We have also added a brief discussion on this when discussing the suggested confusion matrix (see summarizing comment) in the discussion section.

      b. Sec. 4.2 (but also Sec. 4.1). The only way to check the effect of spatial distance between CMLs and disdrometers is to select a subset of CMLs with a disdrometer in the neighbourhood and calculate the MCC:

We have selected disdrometers within 8 km. This was a tradeoff between having enough data and not having disdrometers too far away from the CML. Since we evaluate the RT and CR estimates at similar distances from the disdrometer, the methods should have the same bias. We have experimented with plotting the MCC for different CML-disdrometer distances,

and we find that the MCC only decays slightly with distance within this range. Moreover, the CR method really only provides an improvement at temperatures around 0 degrees, making temperature a more interesting variable. We suggest not to show a figure of the metrics for different distances, as it does not directly contribute to better understanding the difference between the CR and RT method.

      c. About wet snow. Looking at Fig. 4 wet snow seems to correlate with Td (as expected). I expect wet snow to occur at small negative values of Td, while dry snow to occur at colder temperatures. Below -5°C it's mostly dry snow according to Fig. 4. Can the author refine a little bit their decision algorithm including a Td threshold to discriminate between wet and dry snow? Could it be that some unexpected results are due to the way data with different sampling rates were combined together in the hourly time windows and the threshold values used to flag rain/snow /mix/dry intervals)

That would indeed be interesting. Unfortunately the disdrometers do not provide estimates about dry and wet snow, making such an experiment difficult to do. In the above comments we have suggested simplifying the classification method so that the assumptions made are more explicit. This would still allow for a discussion about the effect of wet snow, but more related to the performance of the CR and RT method. In the revised manuscript we provide an extensive discussion on the effect of mixed precipitation (L340-L377).

**Minor comments:**
Line 44: "Human observations can be subjective and aren't suitable for continuous high frequency monitoring" The term "high frequency" is a bit ambiguous in this context. I guess you mean high-rate monitoring.

In the revised manuscript we have changed this line to read (L44):
- "Human observations can be subjective and aren't suitable for continuous high-rate monitoring."

Lines 174-175: I cannot understand the statement. "Our dataset consists of CML-disdrometer pairs from the summer dataset and the winter dataset. Every minute each pair provides several different observations such as disdrometer observed precipitation type, dew point temperature and CML signal loss.". Isn't the dew point temperature taken from the ERA5 dataset? "Our dataset" in my eyes are the data "we produced ourselves", but it's not the case. Why radar maps are not part of "our dataset"?

In the revised manuscript we have removed these lines as we agree that they are confusing.

Line 174: you state that disdrometer data are taken every 1-min while previously (line 128) you stated that precipitation type classification from disdrometers is available every 10-min. Please clarify.

In the revised manuscript we have simplified the aggregation method so that it is easier for the reader to understand all figures, see major points 2 and 3.

Lines 179-182, comment on second row of the figure. I would say "snow mostly below 0°C" while "rainfall is mainly above 0°C".

Yes. In the revised manuscript we have changed this Figure to include the RT and CR estimates instead.

Line 186: "using monthly time series". To me "monthly time series" means that you have used a time series long one month and extract information from the time series as a whole. Please explain-

In the revised manuscript we have deleted this line as it does not convey any important information.

Line 187: pair instead of pairs

In the revised manuscript we have changed the word to "pair".

Figure 2: the gray cluster is hardly visible in the two bottom panels. It would be better to use a darker tone of gray.

We in the revised manuscript we have changed the color map to one of matplotlibs sequential colormaps

Lines 186-192 (Comments of Figure 2): I see MCC_rain of CR is usually better when it's above 0.4, maybe worth to state it-.

Yes, in the revised manuscript we have added a confusion matrix with corresponding metrics that better highlight this. See Fig. 9 and Table 3 (L187-L188).

Lines 193-196(Comments of Figure 2): the first thing I noticed looking at this figure is that CML really enhances MCC_rain wrt radar (as expected).

That is true. We have added a discussion on this (L331-L338):

Figure 5, caption: I guess it is "mix(red)" instead of "dry(red)".

In the revised manuscript we have removed the mixed class and fixed the caption.

Line 264: 10°C instead

In the revised manuscript we have added ° instead of the circle.

Line 453: DOI is not correct

Ok.

---

## Referee Report (RR1)

**Review of the revised version of the following manuscript**

Title: Combining commercial microwave links and weather radar for classification of dry snow and rainfall
Author(s): Erlend Øydvin et al.
MS No.: egusphere-2024-2625
MS type: Research article

I would like to thank the authors who did a considerable extra effort to reply to the reviewers comments. The new data classification strategy is straightforward and results are presented now in a more clear form.

However, I think some (small) work is still needed.

Major point 1 has been properly addressed.

Major point 2 has been properly addressed after the authors moved to the hourly integration time scale.

Major point 3: I think it has been addressed. I have only an editorial comment on Fig. 7 (same for Fig. 2): I think it would better to use a different colormap on Fig. 7b and Fig. 7c, because it's not easy to distinguish between small and high fractions. For instance, using 5 distinguishable tones of red for snow and of blue for rain (0-20, 20-40,40-60, 60-80 and 80-100) would help in better understanding your statement on lines 259-260: "Additionally, for events where the CML estimates more rainfall 260 (negative bias), there are more rainy hours as observed by the disdrometer."

Major point 4: I do not completely agree with the authors' reply on 4a. It makes sense to feed the RT method with ERA5 re-analysis data and indeed I didn't ask the authors to feed the RT method with ground data. I think it would be useful to check ERA5 against some available ground data. The authors discuss in depth about possible errors of disdrometer but do not talk about possible errors in Td estimates.

Comments on the new version:
- Fig. 6 c) deserves an explanation. Why CR method detects a lot of snow events with Td well above 0°C (there is a peak around 8-9° C!). This is an important fact spotted on lines 250-251 and quantified in Figure 9f). However, there is only a short comment on lines 324-326 later on in the discussion. Is this really an algorithmic problem, as the authors suggest (hence, it is anything that can be mitigated) or, rather, is it due to an inner limitation of CML sensors, i.e. a limited sensitivity due to length-frequency characteristics + signal quantization? Unfortunately, there is not an analysis divided by rainfall intensity class, which I think would help in understanding. At least, please add this possible explanation related to a limited sensitivity to the CML.

- Fig. 2 caption: what do the author mean by "(distorted) positions of the CMLs" ?

- Lines 230-236 the description of what's happening around 18:00 does not fit Fig. 3 ("Around 18:00, the CR method, RT method, and disdrometer all estimate rainfall")

   Conclusions/1: Just counting the misclassified samples on Fig. 9 (i.e. non-diagonal elements of the matrices), for Td>=2°C I see roughly 35k misclassified hours by RT and 39k by CR, while if Td<=-2°C we have 30k against 32k respectively and if |Td|<=2°C we have 22k against 21k. Either method has its own strengths and weaknesses, which brings to the next bullet.

- Conclusions/2: I would like that the authors comment about whether merging RT + CR methods would be beneficial. Maybe yes, because, for instance, Td info would dump to zero all that huge CR misclassification of dry/rain as snow at large positive Td values, which is the minus of CR.

---

## Author Response (AR2)

We thank the reviewer for the valuable feedback on the revised manuscript. In the new revision, we have incorporated the suggestions from the reviewer and performed a language review.

Please find our reply in blue to the individual issues raised by the reviewer. Note that the line numbering refers to the lines in the new revision.

Review of the revised version of the following manuscript

Title: Combining commercial microwave links and weather radar for classification of dry snow and rainfall
Author(s): Erlend Øydvin et al.
MS No.: egusphere-2024-2625
MS type: Research article

I would like to thank the authors who did a considerable extra effort to reply to the reviewers comments. The new data classification strategy is straightforward and results are presented now in a more clear form.

However, I think some (small) work is still needed.

Major point 1 has been properly addressed.

Major point 2 has been properly addressed after the authors moved to the hourly integration time scale.

Major point 3: I think it has been addressed. I have only an editorial comment on Fig. 7 (same for Fig. 2): I think it would better to use a different colormap on Fig. 7b and Fig. 7c, because it's not easy to distinguish between small and high fractions. For instance, using 5 distinguishable tones of red for snow and of blue for rain (0-20, 20-40,40-60, 60-80 and 80-100) would help in better understanding your statement on lines 259-260: "Additionally, for events where the CML estimates more rainfall 260 (negative bias), there are more rainy hours as observed by the disdrometer."
Good idea. In the revised manuscript we have changed the colormap in Fig. 2 and Fig. 7 to use a colormap that divides the colors into distinguishable tones.

Major point 4: I do not completely agree with the authors' reply on 4a. It makes sense to feed the RT method with ERA5 re-analysis data and indeed I didn't ask the authors to feed the RT method with ground data. I think it would be useful to check ERA5 against some available ground data. The authors discuss in depth about possible errors of disdrometer but do not talk about possible errors in Td estimates.
We agree that we should discuss the uncertainties of the temperature data. The temperature data is a downscaled version of ERA5 temperature data that is combined with ground observations of temperature. This information is included in the sources, but it was not made explicitly clear in the previous version of the manuscript. Lussana et al., (2019) perform an extensive analysis of this dataset with some case studies. To better highlight the uncertainties of the temperature data we suggest the following changes to the manuscript:

Add the following to the methods section (L147):
- "Temperature and humidity data were downloaded from THREDDS (2024). The temperature data is a downscaled version of ERA5 data that is combined with ground observations on a 1 km grid with a temporal resolution of 1 hour (MET, 2024; Lussana et al., 2021, 2019). Lussana et al., (2019) provide an extensive analysis of this data, and the uncertainty of the data depends on several factors like distance to the closest observation station, terrain complexity and model assumptions.

Add the following to the discussion (L375):
- "Another source of uncertainty lies in the temperature data used for the RT method. The temperature data is a downscaled version of ERA5 data that is combined with ground observations. Lussana et al., (2019) found that the expected RMSE of the temperature data ranged between 1-2 °C in observation dense areas and 2-2.5 °C in observation sparse regions. The RT method performance could thus be less good in areas with complex terrain and sparse ground observations."

Comments on the new version:
• Fig. 6 c) deserves an explanation. Why CR method detects a lot of snow events with Td well above 0°C (there is a peak around 8-9° C!). This is an important fact spotted on lines 250-251 and quantified in Figure 9f). However, there is only a short comment on lines 324-326 later on in the discussion. Is this really an algorithmic problem, as the authors suggest (hence, it is anything that can be mitigated) or, rather, is it due to an inner limitation of CML sensors, i.e. a limited sensitivity due to length-frequency characteristics + signal quantization? Unfortunately, there is not an analysis divided by rainfall intensity class, which I think would help in understanding. At least, please add this possible explanation related to a limited sensitivity to the CML.
Good point. There are several things that could affect the CML. We suggest adding the following to L322:
- "The large number of false snow events estimated by the CR method, also observable above 2 °C (Fig. 6), might be due to several factors. Low intensity rainfall events could fail in triggering the CML rainfall detection algorithm, for instance due to the quantization of the CML signal. Further, due to the spatial difference between the radar beam and the CMLs, the precipitation might hit the radar, but miss the CML, triggering the CR method to estimate snow. Finally, hardware issues with the CML, or database errors, could result in a flat signal level, causing the CR method to misinterpret conditions and estimate snow. Better quality control of the CMLs, for instance by checking their correlation against the weather radar during rainfall events could improve the CR estimates."

• Fig. 2 caption: what do the author mean by "(distorted) positions of the CMLs" ?
We suggest rephrasing the line to read:
- "White circles indicate the distorted positions (i.e. randomly shifted position to prevent exact retrieval of coordinates) of the CMLs."

• Lines 230-236 the description of what's happening around 18:00 does not fit Fig. 3 ("Around 18:00, the CR method, RT method, and disdrometer all estimate rainfall")

It's true that the RT method also estimates some snow. For making the text simple to read we suggest deleting this line, as the rest of the text sufficiently well describes the behaviour of CML1 and CML 2.

Conclusions/1: Just counting the misclassified samples on Fig. 9 (i.e. non-diagonal elements of the matrices), for Td>=2°C I see roughly 35k misclassified hours by RT and 39k by CR, while if Td<=-2°C we have 30k against 32k respectively and if |Td|<=2°C we have 22k against 21k. Either method has its own strengths and weaknesses, which brings to the next bullet.

We agree that each method has its own strengths and weaknesses. See also the next comment.

• Conclusions/2: I would like that the authors comment about whether merging RT + CR methods would be beneficial. Maybe yes, because, for instance, Td info would dump to zero all that huge CR misclassification of dry/rain as snow at large positive Td values, which is the minus of CR.

Agree, we suggest to add the following to the discussion (L341):

- "For classifying snowfall and rainfall, both methods have their own strengths and weaknesses. The CR method shows a better ability to classify precipitation in the interval -2 to 2 °C (MCC = 0.39), but falsely estimates a large number of snowfall events above 2 °C. The RT method, on the other hand, provides reliable precipitation classification below -2 degrees (MCC = 0.30) and above 2 °C (MCC = 0.34), while its performance is not as good within the interval -2 to 2 degrees (MCC = 0.28). Consequently, combining the RT and CR method would be optimal. This could, for instance, be done by using the CR method in the interval -2 to 2 °C and the RT method below -2 °C. Above 2 °C, precipitation could be classified as rainfall if either the RT or CR method detects rainfall."